# MOBILEA³GENT: TRAINING MOBILE GUI AGENTS USING DECENTRALIZED SELF-SOURCED DATA FROM DIVERSE USERS

## ABSTRACT

The advancement of mobile GUI agents has opened new opportunities for automating tasks on mobile devices. Training these agents requires large-scale high-quality data, which is prohibitively expensive when relying on human labor. Given the vast population of global mobile phone users, if automated data collection from them becomes feasible, the resulting data volume and the subsequently trained mobile agents could reach unprecedented levels. Nevertheless, two major challenges arise: (1) extracting user instructions without human intervention and (2) utilizing distributed user data while preserving privacy. To tackle these challenges, we propose **MobileA3gent**, a collaborative framework that trains mobile GUI **A**gents using self-sourced data from diverse users. The framework comprises two components, each targeting a specific challenge: (1) **Auto-Annotation**, which enables the automatic collection of high-quality datasets during users' routine phone usage with minimal cost. (2) **FedVLM-A**, which enhances federated VLM training under non-IID distributions by incorporating adapted global aggregation based on both episode-level and step-level variability. Extensive experiments prove that MobileA3gent achieves superior performance over traditional approaches at only 1% of the cost, highlighting its potential for real-world applications. Our code is publicly available at: https://anonymous.4open.science/r/MobileA3gent-Anonymous.

## 1 INTRODUCTION

Mobile GUI agents (Bai et al., 2024; Wang et al., 2024b;a) have experienced significant advancements, propelled by recent progress in Vision-Language Models (VLMs). Designed to simulate human mobile phone usage behavior, mobile agents can automate complex tasks on mobile phones, saving tremendous human labor and change everyday lives. Compared to non-agent solutions, mobile agents offer significantly better adaptability and generalizability, enabling them to effectively handle various mobile environments and operation scenarios (Zhang et al., 2023).

The training of mobile agents heavily depends on large-scale, high-quality datasets (Chai et al., 2024; Song et al., 2024; Zhang et al., 2024c). To build such datasets, existing approaches rely on centralized data collection followed by human annotation, resulting in high costs and limited scalability. To achieve large-scale data acquisition more efficiently, a paradigm shift (as shown in Figure 1) from **centralized** to **distributed** data collection is necessary, enabling diverse users to participate in data contribution. Additionally, replacing **human** annotation with **automatic** annotation is crucial for efficiently processing the vast amount of collected data, allowing direct data sourcing from real user interactions.

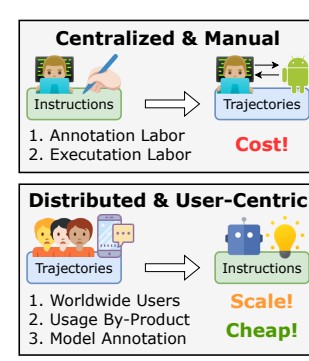

Figure 1: Comparing our proposed paradigm with conventional ones. By leveraging users' daily phone usage, we achieve superior scalability with drastic cost savings.

Our insight is that the frequent and ever-growing phone usage by users worldwide naturally generates valuable supervisory information, which can serve as a rich data source for training mobile agents. Building on this user-centric insight, we aim to effectively utilize these distributed data, while minimizing human involvement in the process. However, two technical challenges remain:

1. Although the users' phone usage provides real-world trajectories (screenshots and actions), it is difficult to extract the real intentions (instructions) behind the actions in natural language;
2. Data collected from one user is both scale-limited and privacy-sensitive. The challenge lies in how to utilize distributed data from diverse users to boost performance while protecting privacy.

To tackle these challenges, we propose **MobileA3gent**, a collaborative learning framework that trains mobile agents using automatically collected user data from daily phone interactions while preserving user privacy. Specifically, MobileA3gent features two novel techniques.

First, we propose **Auto-Annotation**, an automated method for data collection and annotation that leverages locally deployed VLMs to annotate user instructions based on interaction trajectories. The key technical innovation lies in combining step-wise low-level instruction breakdowns with episode-wise summarization, allowing even small local VLMs to better understand the user's intent. The step-wise description decomposes complex user instructions into simpler steps, enabling the VLM to comprehend and extract information more accurately. Meanwhile, the episode-wise summarization provides a global perspective on the entire task, generating a more comprehensive caption of the user's ultimate instruction. Compared with human annotation, Auto-Annotation generates data of comparable quality with minimal cost requirement.

Second, to effectively utilize decentralized data from diverse users, we propose **FedVLM-A**, which pioneers the integration of Federated Learning (FL) (Kairouz et al., 2021) and collaborative training of VLM-based GUI agents, while ensuring rigorous user privacy protection. We further propose a novel aggregation method, termed **A**dapted global aggregation, which accounts for both episode-level and step-level distributions to handle the two-level heterogeneity (formulated in Section C) in diverse users' data, overcoming the limitations of traditional one-level aggregation methods (Karimireddy et al., 2021; McMahan et al., 2017; Hsu et al., 2019; Reddi et al., 2020). Adapted aggregation adapts the global aggregation weights using a weighted sum of episode and step counts for each client, thereby enhancing the performance of mobile agents trained in non-IID scenarios.

Extensive experiments on four benchmarks with 10+ models and metrics demonstrate that: (1) MobileA3gent achieves the best all-around trade-off across four dimensions, delivering performance on par with centralized manual approaches at significantly lower cost, while also ensuring privacy and achieving exceptional scalability. (2) Auto-Annotation outperforms all annotation baselines in performance while reducing annotation costs by 99% compared to manual labeling. (3) FedVLM-A achieves an at least 5% relative improvement over representative FL baselines in non-IID scenarios. These promising results underscore the immense potential of our framework to serve as a novel and practical paradigm for real-world applications. To summarize, our contributions are as follows:

1. We formulate the problem of self-sourced data collection from distributed mobile phone users and propose Auto-Annotation, an automatic data collection method, which achieves data quality comparable to human-annotated data at a significantly lower cost.
2. We introduce MobileA3gent, a collaborative learning framework for training mobile agents on decentralized user data while preserving privacy. By incorporating FedVLM-A, we enable federated training of VLMs and achieve superior performance when confronted with heterogeneity.
3. We conduct extensive experiments across comprehensive benchmarks and metrics. The compelling results highlight the substantial potential of our approach for real-world applications.

## 2 PROBLEM FORMULATION

### 2.1 PRELIMINARIES

**Data Composition.** The mobile GUI agent, powered by a VLM, simulates human users and completes tasks in a step-wise process. To train the core VLM, one data episode, denoted as $\mathcal{D}$, comprises multiple steps, each serving as a basic training unit. A step consists of three components: a task instruction $\mathcal{T}$, a screenshot, and a corresponding action. The composition of a data episode is defined as: $\mathcal{D} = \{\langle \mathcal{T}, a_i, s_i \rangle \mid i \in [1, n]\}$, where $\langle \mathcal{T}, a_i, s_i \rangle$ represents the $i$-th step, with $a_i$ and $s_i$ denoting the action and screenshot respectively.

**Traditional Approach.** Automating mobile devices poses significant challenges, leading to a heavy reliance on high-accuracy training data, which are, at present, almost all annotated by humans. The traditional paradigm (Li et al., 2024b; Qin et al., 2025; Hong et al., 2023) thus involves: (1) manually authored task instructions, followed by (2) centralized data collection and model training.

As shown in Figure 1, this approach typically outsources instruction writing to human annotators using predefined rules or heuristics to promote both quality and diversity. Each instruction is then executed step-by-step in a controlled environment, such as an Android simulator, to collect paired screenshots and actions. To guarantee correctness, all interactions are manually verified, resulting in substantial annotation costs and difficulty in scaling.

## 2.2 PRIMARY PROBLEM

To overcome the high cost and limited scalability of the traditional paradigm, we introduce a novel distributed user-centric approach for training mobile agents. The primary problem we address is: **How to harness private and distributed phone usage trajectories from diverse users?** We further decompose the primary problem into two subordinate problems: (1) How to automatically collect data from individual users without incurring expensive human annotation; and (2) How to effectively utilize decentralized data to optimize the agent while preserving user privacy .

**Sub-Problem 1: Automatic Data Annotation on User Side.** During phone interactions, users spontaneously generate screenshots and actions, which are assumed to be easily collectible. However, users do not receive explicit natural language instructions and only act based on their underlying intentions, making task annotation necessary. Since users are generally reluctant to articulate their intentions and such intentions are non-trivial to infer, the first subordinate problem is: underline{how to automatically derive user intentions without human intervention}, thereby constructing the training dataset. The objective is to learn a function $f(\cdot)$ that predicts user intention $\mathcal{T}^*$, an approximation of task instruction $\mathcal{T}$, based on $n$ steps of actions and screenshots $\langle a_i, s_i \rangle$, that is: $\mathcal{T}^* = f(\{\langle a_i, s_i \rangle\}_{i=1}^n)$ .

**Sub-Problem 2: Distributed Training of Mobile GUI Agents.** The daily phone usage of an individual generates a limited dataset, constraining the agent's performance trained solely on it. Fortunately, with millions of users worldwide, there is immense potential to collaboratively train a mobile agent using their combined data, enabling virtually unlimited scalability. Nevertheless, directly sharing user data poses significant privacy risks, necessitating its use in a distributed manner. Therefore the second subordinate problem is: underline{how to conduct privacy-preserving collaborative training of mobile agents on distributed user data}.

## 3 METHODOLOGY

### 3.1 MOBILEA3GENT: SYSTEM OVERVIEW

**MobileA3gent** is a collaborative learning framework for training mobile GUI agents that automatically annotates user instructions and enables privacy-preserving utilization of distributed user data. It consists of two components to address the two subordinate problems in Section 2.2 respectively.

(1) First, as shown on the left of Figure 2, to mitigate the high cost produced by human annotation, we propose **Auto-Annotation** to automatically construct datasets for each participating user. By leveraging trajectories as by-products of users' daily phone interactions, we eliminate the need for centralized, manual task execution. Since the actual user instructions are not directly observable during usage, a local VLM is deployed to infer them automatically, significantly reducing annotation costs. Inspired by the human reasoning process, we decompose the entire task into multiple steps, allowing the VLM to better interpret fine-grained intentions. (2) Second, as depicted on the right side of Figure 2, diverse users collaborate via **FedVLM-A** to jointly train a target mobile agent with enhanced capabilities. Federated learning is adopted to facilitate effective collaboration while ensuring privacy protection. Through the integration of FL and VLM training, each user trains a local model on auto-annotated data and uploads it to the server. The server aggregates uploaded models using an adapted global aggregation method, which improves traditional methods in heterogeneous scenarios by considering both episode-level and step-level distributions.

### 3.2 AUTO-ANNOTATION: AUTOMATIC DATA ANNOTATION FROM DAILY PHONE USAGE

Auto-Annotation functions by automatically building datasets from users' daily phone usage without manual effort. Screenshots and actions are directly recorded from user trajectories. To annotate user instructions, the idea is to employ a local annotation model to progressively decode user intent in a step-by-step manner, which comprises three stages: (1) Converting coordinate-based actions into semantically meaningful descriptions; (2) Incrementally generating low-level instructions to reflect each discrete operation; (3) Consolidating these atomic instructions into a high-level instruction for

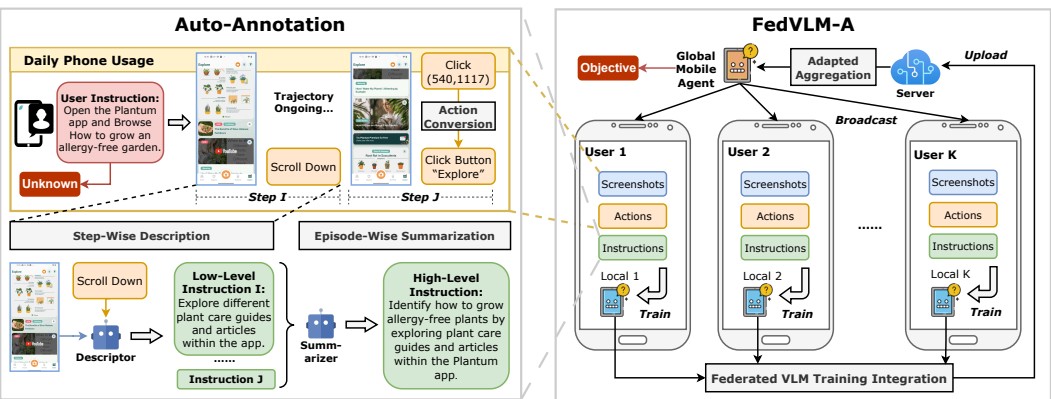

Figure 2: System overview of *MobileA3gent*. During individual users' daily phone usage, Auto-Annotation automatically constructs training data through step-wise description and episode-wide summarization. Each user then participates in FedVLM-A through our training integration. By applying adapted global aggregation, we obtain the target mobile agent with enhanced capabilities.

the entire episode. Note: A low-level instruction is a specific, atomic directive that corresponds to an individual step, whereas a high-level instruction represents the overall task objective.

**Rule-Based Action Conversion.** As indicated by previous works (Zheng et al., 2024), some VLMs, such as GPT-4V, are unable to effectively identify the location of operations. Therefore, to make the original actions interpretable to the local annotation model, we adopt a rule-based technique rather than using models (Wang et al., 2024a) to transform the action into a natural language sentence. Specifically, for `CLICK` actions, we align the exact click position with a corresponding interface element based on the accessibility tree. If the element contains text or invokes a function, we use the associated text or function name to construct a meaningful action description. For other actions, such as `NAVIGATE_HOME`, we slightly adjust the phrasing to improve clarity and readability. A code snippet is included in Appendix F.

**Step-Wise Instruction Description.** During this stage, we annotate users' low-level atomic instructions through step-wise description, a novel technique that decomposes complex user tasks into multiple steps. Specifically, at each step $i$, the local annotation model $\mathcal{M}_a$, referred to as the *Descriptor*, is prompted to generate an atomic instruction that reflects the user's explicit intent, as:

$$\text{Descriptor} : \langle s_i, A_i \rangle \xrightarrow{\mathcal{M}_a} \mathcal{T}_i^{\text{low}}, \tag{1}$$

where $\mathcal{T}_i^{\text{low}}$ is the prediction of user intention, serving as an approximation of the actual low-level instruction. $s_i$ and $A_i$ respectively represent the current screenshot and the corresponding converted action. For the example in Figure 2, the atomic intent of `Scroll Down` on the browsing page is to "explore more articles on plant care". When combined with rule-based action conversion, the step-wise description allows the model to focus on localized context at each interaction, leading to more accurate and interpretable low-level directives. This step-by-step procedure also ensures that the information is more finely processed, facilitating better high-level summarization in subsequent stages Details of our prompt templates can be found in Appendix F.6.

**Episode-Wise Intention Summarization.** This stage generates high-level instructions by summarizing the low-level instructions from all steps. The novelty lies in providing global context enriched with step-wise details, enabling the annotation model to effectively extract user intention. To provide global visual context for the annotation model $\mathcal{M}_a$, referred to as the *Summarizer* in this step, we concatenate all relevant screenshots into a single image $s_c$, arranged in chronological order. Note that this approach (1) allows *Summarizer* to develop a comprehensive understanding of the entire task sequence, and (2) eliminates the need for multiple inferences by performing inference only once. Finally, we compile the concatenated screenshot $s_c$ and the list of low-level instructions $\{\mathcal{T}_i^{\text{low}}\}_{i=1}^n$ into a single prompt and feed it into $\mathcal{M}_a$ to summarize the user's overall intention $\mathcal{T}^{\text{high}}$ as:

$$\text{Summarizer} : \langle s_c, \{\mathcal{T}_i^{\text{low}}\}_{i=1}^n \rangle \xrightarrow{\mathcal{M}_a} \mathcal{T}^{\text{high}}. \tag{2}$$

Since users give no explicit commands, $\mathcal{T}^{\text{high}}$ simulates what they would convey if asking an agent to perform the same task. Combined with above mentioned techniques, episode-wise summarization produces high-quality instructions comparable to human annotated data, all while exclusively using locally deployed VLMs, thereby substantially reducing costs.

### 3.3 FedVLM-A: Federated VLM Agents with Adapted Global Aggregation

To facilitate training mobile GUI agents on distributed user data without comprising privacy, we propose FedVLM-A, a novel collaborative framework which pioneers the integration of federated learning with VLMs and improve performance in heterogeneous scenarios with Adapted aggregation.

**Integrating VLM Training.** We build upon the highly-starred training framework, ms-swift (Zhao et al., 2024), and successfully extend it to support federated VLM training. We ensure the algorithmic correctness by following the implementation of federated training frameworks for Large Language Models (LLMs) (Ye et al., 2024). To enhance training efficiency and better accommodate user-side resource constraints, we incorporate Low-Rank Adaptation (LoRA) (Hu et al., 2021). In our federated setting, $K$ clients (users) collaborate with a central server to train a global VLM without directly sharing private data. At each communication round $l$, the server broadcasts the global model $\mathcal{M}^{(l)}$ to all participating clients $u_k \in \mathcal{S}^l$, who initialize their local models accordingly: $\mathcal{M}_k^{(l,r+1)} := \mathcal{M}^{(l)}$, where $\mathcal{M}_k^{(l,0)}$ denotes the local model at the $l$-th round and 0-th training iteration. Each client $u_k$ then conducts multiple iterations of stochastic gradient descent (SGD) updates on its local dataset $\mathcal{D}_k$. At each iteration $r$, with learning rate $\eta$, the local model is updated as follows:

$$\mathcal{M}_k^{(l,r+1)} = \mathcal{M}_k^{(l,r)} - \eta \nabla \ell(\mathcal{M}_k^{(l,\tau_k)}); \mathcal{T}, s, a), \tag{3}$$

where $\ell(.)$ represents the computed loss based on a data sample $\langle \mathcal{T}, s, a \rangle$.

**Adapted Global Aggregation.** In this stage, the server updates global model by aggregating local models, which is subsequently broadcast to available clients for the next round. Our innovation lies in adapting the aggregation strategy to accommodate the two-level structure of datasets used for training mobile agents, encompassing both step-level variations and episode-level distributions. Traditional FL methods use the sample number of client as the aggregation weight. This insight has been proven successful over the past several years (Li et al., 2019; 2023). However, prior aggregation methods, such as FedAvgM and FedYogi (McMahan et al., 2017; Hsu et al., 2019; Reddi et al., 2020), which perform well on tasks such as image classification, overlook the two-level distribution discussed in Section C. These methods treat all samples equally, regardless of whether they originate from the same episode or not, thereby ignoring structural dependencies.

To address this limitation, we propose a novel aggregation technique adapting to the new scenario of MobileA3gent. Within federated training of mobile agents, the data samples can be measured by both step count $n_k$ and episode count $n_k^{epi}$. $n_k^{epi}$ is as well as, or even more important as it indicates how many tasks the agent has learned on. As $n_k^{epi}$ and $n_k$ are measured in different scales, we empirically set a hyper-parameter $\lambda$ to align them, which is calculated around the average step length of all episodes. Then we redefine the sample count as $n_k^*$ and reformulate the aggregation weight based on our adapted sample count $n_k^*$; that is:

$$n_k^* := \lambda n_k^{epi} + n_k; \quad \omega_k = \frac{n_k^*}{\sum_{k \in \mathcal{S}^l} n_k^*}, \tag{4}$$

where $\omega_k$ denotes the weight for client $u_k$ and $\mathcal{S}^l$ is the sampled participating clients. This design smoothly improves upon traditional aggregation and inherits its convergence property. When $\lambda = 0$, it degrades to normal aggregation. Finally, the global model $\mathcal{M}^{(l)}$ is adaptively aggregated as:

$$\mathcal{M}^{(l+1)} := \sum_{k \in \mathcal{S}^l} \omega_k \mathcal{M}_k^{(l)}. \tag{5}$$

| Privacy Protection | Eavesdrop | Abuse | Exposure |
|---|:---:|:---:|:---:|
| API-Based Agent | ✗ | ✗ | ✗ |
| DistRL* | ✓ | ✗ | ✗ |
| MobileA3gent | ✓ | ✓ | ✓ |

Table 1: Comparing privacy protection against risks. FedVLM-A offers strongest protection by addressing all three identified concerns. In contrast, *API-Based Agent* directly transmits user data, while *DistRL* stores all data centrally for training.

The adapted aggregation in FedVLM-A balances both episode and step counts, achieving a better utilization of decentralized data from heterogeneous users.

**Secure Aggregation.** FedVLM-A preserves privacy by keeping original user data, which may contain sensitive information, on users' local devices without transmitting. Furthermore, we adopt secure aggregation (Bonawitz et al., 2016) by injecting a pre-generated random gradient into the uploaded LoRA adapters. Specifically, participating clients are randomly grouped into pairs; within each pair, the two clients respectively add and subtract the same random gradient. This pairing strategy ensures that the injected randomness cancels out during aggregation, leaving the final global model unaffected.

| Methodology | AndroidControl-High | | | AndroidControl-Low | | | Anno. Cost | Privacy Protect | Scalability |
|---|---|---|---|---|---|---|---|---|---|
| | Type | Ground | SR | Type | Ground | SR | | | |
| *Prompting using Open-Ended & Closed-Ended Models* | | | | | | | | | |
| OS-Atlas-7B (Wu et al., 2024) | 57.44 | 54.90 | 29.83 | 73.00 | 73.37 | 50.94 | - | ✓ | No |
| GPT-4o (OpenAI, 2023) | 66.17 | 3.38 | 16.69 | 87.03 | 6.06 | 31.15 | - | ✗ | |
| *Finetuning on Human-Annotated Data* | | | | | | | | | |
| Central-Human (Li et al., 2024a) | 74.41 | **53.75** | 50.97 | **97.02** | 74.66 | 80.40 | 10880 | ✓ | Very Low |
| FedL/VLM (Ye et al., 2024) | 68.55 | 36.90 | 39.79 | 95.38 | 56.30 | 69.00 | | | |
| *Finetuning on Synthetic Data* | | | | | | | | | |
| OS-Genesis (Sun et al., 2024) | 66.15 | - | 44.54 | 90.72 | - | 74.17 | $\approx 10^3$ | ✓ | Limited |
| *Finetuning on Auto-Annotated User data* | | | | | | | | | |
| DistRL* (Wang et al., 2024d) | 73.62 | 51.14 | 48.58 | 96.42 | 75.13 | 80.18 | 152.92 | ✗ | Very High |
| MobileA3gent | **74.66** | 53.05 | **57.24** | **97.17** | **76.98** | **81.52** | | ✓ | |

Table 2: Multi-dimensional comparison of MobileA3gent with other approaches. With 1% overall cost, MobileA3gent even surpasses the centralized human-annotated data. * We adjust DistRL to our user-centric setup. Anno. Cost refers to annotation cost in terms of cents (¢). Colors indicate preferable , moderate and concerning outcomes. Baseline details are explained in Appendix F.5.

## 3.4 PRIVACY ANALYSIS

**Threat Model.** (1) *Adversaries:* we consider external eavesdroppers attempting to intercept communication, honest-but-curious servers that follow protocol but seek to infer private user information from model updates, data-abusive servers that may store or repurpose uploaded data for unintended uses, and curious peer clients aiming to extract information about other users. (2) *Adversary Capabilities:* adversaries may observe communication traffic, inspect uploaded information for gradient inversion attacks to infer screenshots, actions, or user intents, retain or repurpose received data to construct centralized datasets, or identify client-specific patterns. (3) *Trust Assumptions:* client devices behave correctly without intentional data exfiltration, the server does not tamper with models but may analyze updates, communication channels remain encrypted (e.g., TLS) though passively monitorable, and attackers cannot directly access raw screenshots or logs stored on devices.

**Data Exposure and Storage.** *Local-Only Data:* MobileA3gent ensures that raw screenshots, user actions, intermediate auto-annotation artifacts, and locally constructed training episodes remain strictly on-device. *Transmitted Data:* the only information sent to the server consists of highly compressed LoRA parameter updates, which contain no raw textual or visual user data.

**Privacy Guarantees.** As shown in Table 1, MobileA3gent offers the following privacy protection. *(1) Local Data Retention:* all sensitive GUI content, behavior logs, and annotated intent signals remain on-device, preventing any raw data leakage. *(2) Resistance to Eavesdropping:* only encrypted LoRA updates are transmitted, ensuring that network listeners cannot infer screenshots, actions, or personal information. *(3) Limited Server Inference:* the server receives only sparse LoRA updates and minimal metadata, making gradient inversion significantly harder and preventing reconstruction of user intents or application states. *(4) Prevention of Data Abuse:* as no content-level data are uploaded, the server cannot repurpose, store, or resell user data, and no centralized dataset exists that could be leaked. *(5) Peer Isolation:* clients never interact with one another directly, and all aggregation removes client-specific details, preventing users from inferring or accessing other participant's data.

## 4 EXPERIMENTS

### 4.1 BASIC SETUPS (*More Details in Appendix F*)

**Models, Datasets & Benchmarks.** The base model for most experiments is Qwen2-VL-Instruct-7B (Wang et al., 2024c). We also compare results with 10+ representative models, e.g. InternVL2 (Chen et al., 2024b) in Section 4.5. We select totally three offline agent benchmarks: AndroidControl (Li et al., 2024a), Android in the Wild (AitW) (Rawles et al., 2023) and GUI Odyssey (Lu et al., 2024b). These datasets are collected by crowdsourcing and serve well as a simulation of real-world mobile data. Most experiments are conducted using AndroidControl. Additionally, we employ AndroidWorld (AW) (Rawles et al., 2024), a challenging online benchmark running on Android emulators.

**Performance Metrics.** Following previous works (Wu et al., 2024; Sun et al., 2024; Qin et al., 2025), we utilize three commonly used metrics for GUI agents that assess the accuracy of action

type prediction, coordinate prediction, and step success rate, denoted as *Type*, *Ground*, and *SR*, respectively. *Type* measures the exact match score between the predicted action types and the ground truth. *Ground* evaluates the performance of GUI grounding in downstream tasks. *SR* is short for the step-wise success rate, where a step is deemed successful only if both the action type and its associated arguments are correct. We assess data quality by measuring the similarity between our generated instructions and the ground truth from the original datasets. A comprehensive set of metrics are employed, including text-based metrics such as ROUGE, BLEU, and METEOR (Lin, 2004; Papineni et al., 2002; Banerjee & Lavie, 2005), as well as embedding-based metrics using two of the most downloaded embedding models on Hugging Face, i.e., `jina-v3` and `mxbai-v1`.

**Efficiency Metrics.** We also compare the annotation costs across methods to assess efficiency. The cost of a single human-annotated sample is derived from a Refuel-AI technical report. The costs for model-annotated samples are estimated by calculating the average GPU usage during generation, given by: Anno. Cost $= \left(\frac{\text{Price}}{3600}\right) \times \text{Time} \times \frac{\text{Memory}_{\text{Use}}}{\text{Memory}_{\text{Total}}}$ , where Price is the GPU rent per hour, approximately \$0.2857 for one RTX 4090 GPU we use. $\text{Memory}_{\text{Use}}$ and $\text{Memory}_{\text{Total}}$ represent the average occupied GPU memory and the total memory of the system, respectively. Time is the generation duration measured in seconds. All cost numbers are presented in terms of cents ($\cent$).

## 4.2 Overall Evaluation of MobileA3gent

**Baselines.** To collect data and train mobile GUI agents, we compare *MobileA3gent* against the following baselines: (1) *Central-Human* (Li et al., 2024a), the conventional approach that relies on human annotation and centralized training on a server. (2) *FedLLM/VLM* (Ye et al., 2024), which differs from *Central-Human* by training in a distributed manner across client devices. (3) *OS-Genesis* (Sun et al., 2024), which automates synthetic data generation to reduce human effort. (4) *DistRL\** (Wang et al., 2024d), an adapted version of the original method that first collects decentralized user data and then performs centralized training. During federated training, we randomly select 30% of clients in each round to mimic real-world scenarios where **users are intermittently offline** (Jiang et al., 2024). We evaluate the models at round 30, which corresponds to an expected cumulative client participation of 90%. We also provide prompt-based baselines, using locally deployed models or closed-ended models accessed via APIs, for reference.

**Results & Analysis.** As demonstrated in Table 2, we evaluate from four dimensions: *Performance*, *Efficiency*, *Privacy*, and *Scalability*, and summarize the following key findings: (1) **Comparable performance to *Central-Human*.** As the number of participating clients and the data volume increase, the performance of the collaboratively trained global model via MobileA3gent improves accordingly. Once the participation exceeds a certain threshold, users can obtain a highly capable mobile agent, comparable to or even surpassing *Central-Human* at minimal costs. (2) **Most efficient by leveraging daily phone usage.** The per-client annotation cost remains nearly negligible compared to *Central-Human*. Although *OS-Genesis* also aims to reduce human labor, it first generates synthetic instructions and then collects trajectories by employing GPT-4o to perform tasks in simulators, which still incurs medium-level costs. In contrast, we directly collect trajectories from users' daily phone usage by merely recording interactions, offering the most cost-saving approach for constructing GUI agent datasets. (3) ***MobileA3gent* substantially reduces privacy risks,** by keeping data on local devices. The privacy protection level is comparable to that of locally deployed agents, while achieving significantly higher performance. (4) **Promising scalability based on worldwide users.** As shown in Figure 7, the mobile user base is massive and continually expanding, which enables *MobileA3gent* to achieve much greater scalability compared to other approaches.

## 4.3 Data Quality and Training Evaluation of Auto-Annotation

**Baselines.** For annotating user instructions based on available information, we compare four baselines, arranged from simple to complex. An illustrative example is shown in Figure 11, and the prompts are detailed in Appendix F.6. (1) *Action-Origin*, which concatenates the original formatted action into a string without inference. (2) *Visual-Sense*, which provides the concatenated screenshot to the annotation model for inference. (3) *Self-Instruct* (Wang et al., 2023), which provides all actions simultaneously for the annotation model to infer the intention. (4) *Chain-of-Thought* (Berkovitch et al., 2024), which, at each step, predicts the current intention based on all previous information, with the final intention produced after the task sequence is completed. (5) *Human-Annotation*, which uses the human annotated gold instructions from the datasets. Note that all six methods, including ours, differ only in the instructions used during training. We also include few-shot baselines on widely

| Methodology | AndroidControl-High | | | AndroidControl-Low | | | GUI Odyssey | | |
|---|---|---|---|---|---|---|---|---|---|
| | Type | Ground | SR | Type | Ground | SR | Type | Ground | SR |
| Qwen2-VL-7B-Instruct | 28.61 | 0.00 | 1.94 | 71.09 | 2.19 | 6.41 | 2.87 | 2.94 | 0.51 |
| GPT-4o | 66.17 | 3.38 | 16.69 | 87.03 | 6.06 | 31.15 | 37.50 | 14.17 | 5.36 |
| OS-Atlas-7B-Pro (Wu et al., 2024) | 57.44 | 54.90 | 29.83 | 73.00 | 73.37 | 50.94 | 60.42 | 39.74 | 26.96 |
| Human-Annotation (Li et al., 2024a) | 75.41 | 53.75 | 50.97 | 97.02 | 74.66 | 80.48 | 78.85 | 64.92 | 55.22 |
| Action-Origin | 65.28 | 3.18 | 9.84 | 94.19 | 3.56 | 26.68 | 62.36 | 13.32 | 14.33 |
| Visual-Sense (Zhang et al., 2024a) | 77.50 | 61.13 | 57.37 | 97.47 | 81.42 | 85.54 | 81.53 | 67.66 | 59.49 |
| Self-Instruct (Wang et al., 2023) | 75.86 | 57.28 | 53.95 | 97.47 | 81.97 | 85.25 | **82.80** | 60.27 | 55.16 |
| Chain-of-Thought (Berkovitch et al., 2024) | **77.94** | 56.96 | 55.89 | 97.17 | 83.20 | 85.39 | 74.37 | 50.80 | 49.33 |
| Auto-Annotation | 77.50 | **62.67** | **58.12** | **98.06** | **83.29** | **86.29** | 81.72 | **69.51** | **60.57** |

Table 3: Training evaluation on AndroidControl and GUI Odyssey with more results in Table 7. We compare our method against various baselines. *Auto-Annotation* achieves superior results across all methods, and shows substantial improvement over *Human-Annotation* with significant cost savings.

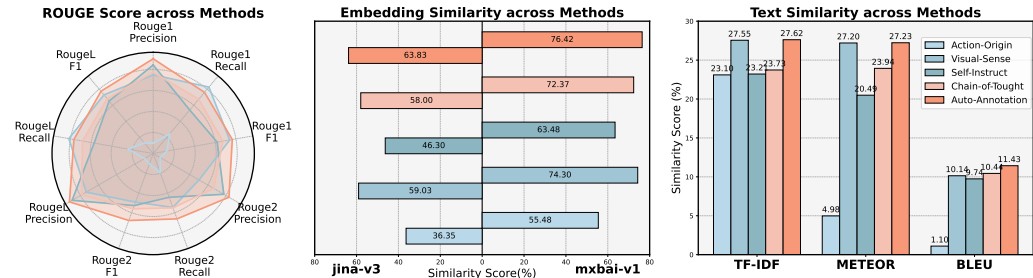

Figure 3: Data quality evaluation across comprehensive metrics. *Auto-Annotation* outperforms all other baselines and achieve comparable quality to *Human-Annotation* with a nearly 80% similarity.

accepted models for reference. For *Human-Annotation*, we report results on 1k samples, whereas other model-based methods use 5k samples for AndroidControl and 3k samples for GUI Odyssey.

**Offline Benchmark.** As shown in Table 3 and Table 8, we summarize the following key findings, (1) **Match or surpass *Human-Annotation*.** Our method achieves performance comparable to human annotation when trained on datasets of equal size. Notably, as the data scale increases, our method surpasses human annotation, highlighting the effectiveness of MobileA3gent and its strong potential for real-world deployment. (2) **Across multiple datasets, our approach consistently outperforms all annotation baselines**, underscoring the robustness and general effectiveness of *Auto-Annotation*. (3) **Drastic cost reductions with minimal accuracy loss.** Combined with the cost statistics in Table 6, By leveraging improved backends such as vLLM, we achieve up to a 99.9% cost reduction with less than a 2% decrease in high-level accuracy. Importantly, even as the dataset size scales up, the cost remains negligible compared to human labor. (4) For the AndroidControl Low-level setting, we observe that all methods achieve comparable performance, particularly for the *Type* metric. This suggests that the low-level task setting is relatively easy for agents to accomplish. (5) **Instruction annotation is critical for agent performance.** Although the data coordinates are identical across all methods, we find that *Action-Origin* still fails on the *Ground* metric. This result underscores the critical importance of incorporating accurate instructions when training GUI agents, thereby highlighting the necessity of annotating user instructions to construct high-quality datasets.

**Online Benchmark.** We further evaluate our approach on the online benchmark AndroidWorld. As shown in Figure 4: (1) *Auto-Annotation* achieves **the best overall performance.** (2) Despite being trained solely on the AndroidControl dataset, the models are able to successfully complete online tasks in a previously unseen environment. This result demonstrates that agents trained with our framework possess **strong generalization capabilities** across unseen tasks and applications. Additional evaluations of generalization performance are provided in Appendix D.5 with Table 9 and 10.

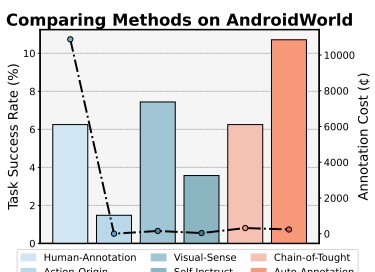

Figure 4: Performance and annotation cost trade-off on AndroidWorld.

**Data Quality.** As shown in Figure 3, (1) *Auto-Annotation* exhibits the **best performance across both text-based and embedding-based metrics**, providing strong evidence for the effectiveness of our proposed hierarchical method. (2) A similarity score of nearly 80% to ground truth further demonstrates the

**practical utility of our generated instructions** on mobile devices, indicating their potential as a viable substitute for human-written ones. (3) *Visual-Sense* delivers competitive data quality using primarily visual signals, suggesting that **even stronger results may be achieved** by integrating *Auto-Annotation* with enhanced visual understanding. We leave this direction for future exploration.

## 4.4 TRAINING EVALUATION OF FEDVLM-A

**Baselines & Splits.** We further conduct experiments under non-IID settings to verify the performance of FedVLM-A and investigate the heterogeneity issue formulated in Section C. We include seven representative FL baselines, such as FedProx (Li et al., 2020), FedYogi (Reddi et al., 2020). To eliminate any potential influence from *Auto-Annotation*, we use the original dataset in this section. Specifically, we sample 1,000 episodes from AndroidControl with uniformly distributed step lengths and create four distinct splits to simulate diverse distribution scenarios. Both the *Step Skew* and

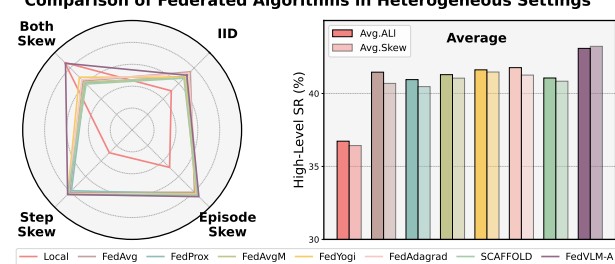

Figure 5: Comparison between FedVLM-A and 7 baselines on non-IID splits of AndroidControl. FedVLM-A achieves SOTA performance on average. Transparent bars indicate average scores over skewed scenarios only.

*IID* splits assign 100 episodes to each client. In the *Step Skew* scenario, clients have an equal number of episodes but varying numbers of steps, whereas in *Episode Skew*, the opposite holds. The *Both Skew* scenario features skewed values for both levels. For baseline *Local*, we evaluate the 0-th client, which, in the *Both Skew* scenario, undergoes a number of iterations comparable to FL baselines.

**Results.** Figure 5 presents the radar chart of all baselines across the four splits, along with the average scores for all scenarios and for the three non-IID subsets. The results reveal the following: (1) Non-IID distributions negatively impact the performance of the global model, underscoring that **data heterogeneity is of critical importance in training distributed mobile agents**. (2) FedVLM-A with adapted aggregation achieves robust performance under non-IID settings, **outperforming all other baselines by at least 5%** in relative improvement. (3) Federated training significantly outperforms local training, validating the benefit of multi-user collaboration. (4) Overall, the results **confirm the existence of the two-level heterogeneity** highlighted in Section C, posing a new challenge for the federated learning community. We hope our findings encourage further exploration in this direction.

## 4.5 ABLATION EXPERIMENTS ON VARIOUS MODELS AND DATA SIZES

**Setups.** We conduct ablation experiments to assess the performance, annotation cost, and time requirements of different base models within *MobileA3gent*. Three configurations are evaluated by varying the choice of annotation and training models, where a combination *x+y* represents using model *x* for annotation and model *y* for training mobile GUI agents. Our model suite includes conversational VLMs such as Phi_3.5 (Abdin et al., 2024), grounding-oriented base models like SeeClick (Cheng et al., 2024) and widely adopted API-based models including GPT-4o/-Mini (OpenAI, 2023). In the plots, icons with light transparency denote models tuned using human annotations, whereas solid icons represent models using *Auto-Annotation*. The horizontal axis reflects annotation cost, measured via the *Pt* backend when applicable, or approximated by model size otherwise. For human-labeled icons, whose actual costs are prohibitively high, we use the same numbers for visualization purposes.

**Results.** As shown in Figure 6, 9 and Table 6, different models exhibit varying trade-offs between performance and cost. We conclude the following observations: (1) **Across all base models, our method achieves consistent improvement** over human-annotated baselines with significant cost savings. (2) The choice of annotation and training models introduces flexible performance–cost trade-offs, allowing practitioners to **tailor configurations to specific deployment constraints**. (3) Within a given VLM family, an increase in parameter numbers generally correlates with higher performance and greater computational demand. While across model types, this correlation does not always hold-e.g., Yi-VL-6B incurs lower costs and performs worse than InterVL2-2B, despite having more parameters. (4) **Qwen2-VL-2B-Instruct (blue circles) achieves the best balance between performance and annotation cost**, making it the most cost-effective option in our study.

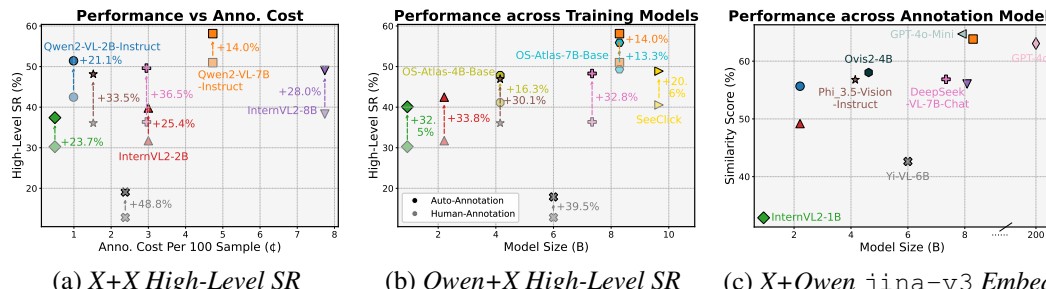

(a) *X+X High-Level SR*  (b) *Qwen+X High-Level SR*  (c) *X+Qwen* `jina-v3` *Embed*

Figure 6: Ablation study on various base models. *x+y* indicates using model *x* for annotation and model *y* for training. *Qwen* refers to the consistent use of Qwen2-VL-7B-Instruct for fair comparison. *X* denotes a specific model. Arrows denote the relative improvement of *Auto-Annotation* over *Human-Annotation*. For the *X+Qwen* setting, we report embedding similarity scores to better capture differences. More comprehensive results are shown in Appendix Figure 9.

| Model | Annotation | | Train | |
|---|---|---|---|---|
| | GPU (MB) | Time/1000 Epi. (s) | GPU (MB) | Time/1000 Epi. (hours) |
| Phi-3.5-Vision-Instruct | 9686 | 1374 | 11558 | 5.21 |
| InternVL2-1B | 3985 | 1662 | 10330 | 3.15 |
| Qwen2-VL-2B-Instruct | 12046 | 1180 | 15680 | 2.58 |

Table 4: Time and computational resources required for training and annotation over 1,000 episodes.

### 4.6 EFFICIENCY ANALYSIS AND RESOURCE STATISTICS

**Computation.** We provide detailed resource statistics, including memory usage and generation time (latency) in Table 4 and 6. The measurements are based on 1,000 samples. For users with only 50 trajectories, both time and cost are significantly reduced, **taking roughly one minute**. Notably, InternVL2-1B has the lowest GPU requirements, needing only 4GB for generation and 10GB for training. All models in Table 4 are **feasible for many mobile devices with 16GB of RAM or more**. Moreover, these requirements can be further reduced by adopting efficient training backbones.

**Communication.** We assume equal episode sizes and compare the communication overhead of three approaches: (1) centralized training (one full dataset transmission), (2) federated training with full-parameter updates, and (3) MobileA3gent, FL with LoRA (transmitting only LoRA adapters). Table 5 demonstrates that MobileA3gent is **the most communication-efficient method**. Regarding frequency, users are expected to interact with the server roughly **once per day**. Since effective training requires only about 10 rounds, this frequency is acceptable and imposes minimal impact on normal device usage.

| Approach | Overhead |
|---|---|
| Central + Unpacked data (10k Epi.) | $\approx$ 100GB |
| Central + Unpacked data (1k Epi.) | $\approx$ 10GB |
| Central + Compressed TFRecord file | $\approx$ 50GB |
| FL + Full model | 16.57GB $\times$ round |
| FL + LoRA adapter (rank=8, $\alpha$=32) | 77.06MB $\times$ round |

Table 5: Communication overhead using Qwen2-VL-7B.

**On-device Deployability.** According to many technical reports (Abdin et al., 2024; Ye et al., 2025), the relative small VLM models including Phi-3.5 and InternVL2-1/2B, are actually deployable on mobile devices. And an increasing number of studies have focused on developing lightweight models tailored for mobile tasks, such as LiMAC (Christianos et al., 2024). These efforts toward building smaller, more efficient VLMs further support the practical viability of on-device GUI agents.

## 5 CONCLUSION

To overcome the scalability and efficiency limitations of traditional mobile agent paradigm, we emphasize the necessity of transitioning from centralized to distributed user-centric data collection, and from human to automatic annotation. To achieve this, we propose MobileA3gent, a framework that collaboratively trains mobile GUI agents using self-sourced data from diverse users. Specifically, we introduce Auto-Annotation, an efficient approach for generating high-quality datasets from routine phone usage at minimal cost. Additionally, we present FedVLM-A, a federated VLM training framework with adapted global aggregation to handle mobile data heterogeneity. Extensive experiments on four benchmarks with 10+ models and metrics validate the effectiveness of MobileA3gent. The promising results highlight the scalability and practicality of our user-centric paradigm, offering a privacy-preserving and cost-efficient solution for training large-scale mobile agent.

## ETHICS STATEMENT

This research focuses on training mobile GUI Agents using self-sourced data from diverse users. The experiments are conducted on public datasets without using actual user datasets, thus introducing no privacy concerns. The datasets are properly cited. No discrimination, bias, or fairness issues are identified in this work.

## REPRODUCIBILITY STATEMENT

To ensure reproducibility, we provide all experimental setups and details in Section 4 and Appendix F.

We have open-sourced our code in the anonymous repository, and will make all data, source code, and model checkpoints publicly available upon acceptance of the paper.

## THE USE OF LARGE LANGUAGE MODELS (LLMS)

We use LLMs to assist with paper writing. Specifically, we first draft the manuscript and then use LLMs (primarily OpenAI GPT models) to improve and revise the text. Afterward, all generated content is manually inspected, and we make final adjustments or rewrites as needed to ensure accuracy.

We additionally employ LLMs to support code writing for simple tasks, such as drawing figures and calculating statistics and use LLMs for repetitive tasks such as reformatting tables into LaTeX. All outputs are double-checked to ensure that no errors are introduced.

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

## A    RELATED WORK

### A.1    DEVELOPMENT OF CURRENT MOBILE GUI AGENTS

The advent of VLMs (Zhang et al., 2024b; Papoudakis et al., 2025; Christianos et al., 2024; Liu et al., 2025a) has marked a significant shift in phone automation, enabling more dynamic, context-aware, and sophisticated interactions with mobile devices (Liu et al., 2025b). Research on mobile agents has progressed through key milestones, with models becoming more proficient at interpreting multi-modal data, understanding user intent, and autonomously executing complex tasks. VLM-based mobile agents typically follow two approaches: (1) Prompt Engineering (Zhang et al., 2023; Lee et al., 2024; Lu et al., 2024d; Chen et al., 2024a), where pre-trained models are guided by carefully designed prompts, and (2) Training-Based Methods (Hong et al., 2023; Cheng et al., 2024), where VLMs are further optimized using large-scale mobile datasets. While training-based methods offer higher potential and generalizability by improving the VLM through fine-tuning, they require a large amount of training data, which can be very costly.

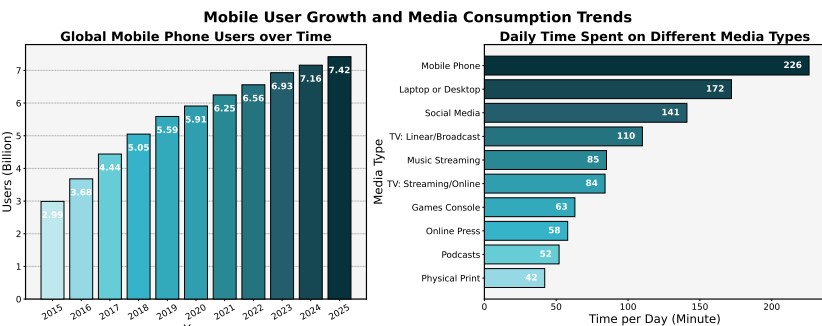

Figure 7: Trends in mobile user statistics. The increasing number of mobile users and their rising daily usage provide a sufficient data foundation for our approach.

## A.2 Efforts in Building Datasets for Mobile GUI Agents

Acquiring training trajectories for mobile agents presents significant challenges. Existing approaches are often reliant on manual curation, making data collection both costly and inefficient. Some works have explored the possibility of automatically constructing datasets using VLMs or Application Programming Interfaces (APIs) (Wang et al., 2021; Lai et al., 2024). But these approaches either halfway to completing the datasets or depend on pre-defined tasks.

OS-Genesis (Sun et al., 2024), the most advanced in this area, proposes reverse task synthesis to eliminate the need for pre-defined instructions. However, this method still requires an agent to execute synthetic tasks in a simulated mobile environment, to obtain the corresponding screenshots and actions. This process does not guarantee the accuracy of executed actions, while also incurs additional computational and resource costs.

In contrast, we propose collecting real-world data from mobile users. This approach offers both (1) unlimited data scale, given the billions of mobile users worldwide, and (2) ground truth accuracy, as the data is directly generated through human execution.

## B Broader Impact and Limitations

### B.1 Broader Impact

In this work, we propose MobileA3gent, a novel framework which can effectively leverage diverse users' daily mobile interactions to train mobile GUI agents without compromising privacy. Our experimental results demonstrate that MobileA3gent holds strong potential for real-world deployment. If successfully adopted, our framework could contribute to society by enabling more capable and personalized agents that automate routine tasks directly on user devices, thereby improving mobile experiences and change everyday life. At the same time, real-world deployment of such systems may introduce additional concerns and risks, particularly related to privacy. Although a thorough investigation of these issues is beyond the scope of this work, we believe our proposed approach lays the groundwork for future research into responsible and privacy-conscious deployment of distributed mobile GUI agents.

### B.2 Limitations

Despite the novelty and promising results of our work, potential limitations remain: (1) Due to device capacity, we are currently unable to conduct experiments on actual user mobile phones, as most mobile phone devices lack the necessary resources to hold mainstream models. However, an increasing number of studies are focusing on developing lightweight models specifically designed for mobile environments (Christianos et al., 2024; Papoudakis et al., 2025). MobileA3gent is model-agnostic and can seamlessly incorporate these smaller, more efficient VLMs, thereby facilitating practical deployment in resource-constrained settings. Also, as shown in Figure 9, we compare models of varying sizes. The results indicate that even compact models—such as InternVL2-1B and Qwen2-2B—can achieve competitive performance with as few as 1,000 training episodes. This demonstrates the scalability and effectiveness of our framework across different model sizes and architectures.

While larger models like Qwen2-VL-7B-Instruct demand more computational resources, the overall annotation cost remains substantially lower than manual labeling, making our approach cost-effective even at scale. Although real-device experiments remain future work, our findings validate the effectiveness of the framework in resource-constrained settings.

## C  DETAILED PROBLEM FORMULATION

In this section, we first briefly elaborate on several key concepts, including the definition of mobile agents, to supplement Section 2.2. We then formulate our federated learning setup, with particular emphasis on the novel heterogeneity introduced by the inherent nature of mobile agent trajectories.

### C.1  SUPPLEMENTAL PRELIMINARIES

**Step-Wise User Phone Usage.** Typically, the process of one user interacting with a mobile device is formulated as follows. Initially, there is a screenshot of the interface, denoted as $s_1$. The user aims to complete a task, denoted as $\mathcal{T}$ in natural language, which requires $n$ steps. Given any screenshot $s_i$, where $i \in [1, n]$, the user performs an action $a_i$, causing the interface to transition from $s_i$ to $s_{i+1}$:

$$\text{User: } s_i \xrightarrow{a_i} s_{i+1} . \tag{6}$$

Once the last action $a_n$ is performed, the interface reaches the final screenshot $s_{n+1}$, finishing the task $\mathcal{T}$ with $n + 1$ screenshots and $n$ actions in total.

**Functionality of Mobile Agents.** The mobile agent, with the core being a VLM denoted as $\mathcal{M}_m$, simulates a human user in a step-wise process for task completion. It operates sequentially when applied to tasks. Given a natural language task $\mathcal{T}$ requiring $n$ steps, at each step $i$, the primary function of the mobile agent is to predict the next action $a_i^*$ required to complete $\mathcal{T}$, based on the current screenshot $s_i$ and contextual information; that is:

$$\text{Mobile Agent: } \langle \mathcal{T}, s_i \rangle \xrightarrow{\mathcal{M}_m} a_i^* . \tag{7}$$

### C.2  FEDERATED LEARNING SETUP

**Reasons for Distributed Training.** The duration of daily mobile phone usage is inherently limited for an individual, resulting in a relatively small dataset collected on a single user's device. This small-scale dataset constrains the performance of the mobile agent trained on it. Fortunately, with millions of mobile users worldwide, there exists a vast opportunity to incentivize users to collaborate and collectively train a mobile agent $\mathcal{M}$, using their combined data. Following the scaling law (Kaplan et al., 2020), leveraging multiple users' data enables virtually unlimited scalability and yields promising results. However, directly sharing or merging data generated from users' daily phone usage poses significant privacy risks. So the local data can only be utilized in a distributed manner.

**Federated Learning.** To address this challenge, we adopt federated learning, which effectively mitigates privacy concerns by keeping data on local devices, and develop a collaborative training framework FedVLM-A for mobile GUI agents. Given the local model $\mathcal{M}_k$ and a data sample $(t, s, a)$ from $\mathcal{D}_k$, the objective of FedVLM-A is to optimize the global model $\mathcal{M}_m$ based on these local datasets; that is:

$$\min_{\mathcal{M}} F(\mathcal{M}) := \frac{1}{K} \sum_{k=1}^{K} \mathbb{E}_{(t,s,a) \sim P_{\mathcal{T} \times \mathcal{S} \times \mathcal{A}}^{(k)}} \left[ \ell(\mathcal{M}_k; t, s, a) \right] . \tag{8}$$

where $\ell : \mathcal{T} \times \mathcal{S} \times \mathcal{A} \to \mathbb{R}_+$ denotes the loss function, e.g. cross-entropy. $P_{\mathcal{T} \times \mathcal{S} \times \mathcal{A}}^{(k)}$ is the distribution over $\mathcal{T} \times \mathcal{S} \times \mathcal{A}$. $\mathcal{T}, \mathcal{S}$, and $\mathcal{A}$ represent task, screenshot, and action spaces, respectively. We assume that the distributions $P_{\mathcal{T} \times \mathcal{S} \times \mathcal{A}}^{(k)}$ differ across clients, which is a common scenario in FL. To the best of our knowledge, we are the first to apply federated learning into the training of mobile agents.

### C.3  NEW HETEROGENEITY

**Two-Level Distribution.** Directly applying federated learning to mobile GUI agents introduces a new form of data heterogeneity. Unlike conventional FL scenarios where data are modeled as flat

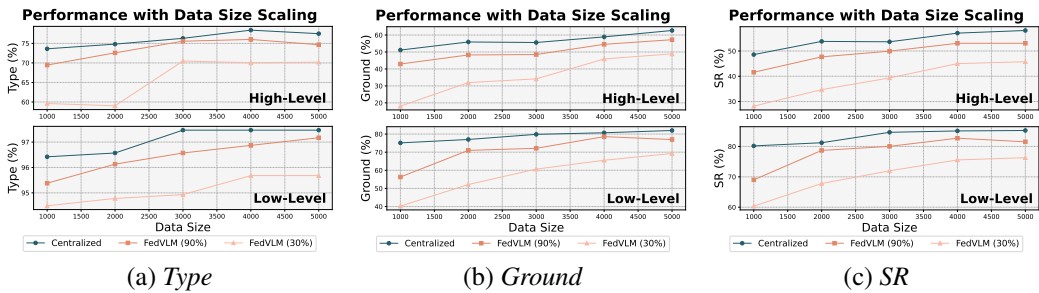

Figure 8: Scaling law analysis on Android Control dataset with different training strategies. All models show a growing tendency with increased data size.

collections of independent samples, mobile interaction data inherently follow a hierarchical structure: they are collected *episode by episode*, with each episode consisting of sequential *steps* governed by a fixed task instruction. As a result, the underlying data distribution operates on two distinct levels. We refer to this structure as the **Two-Level Distribution**.

*Level 1 (Intra-episode)*: Within episode $j$ for user $k$, the task instruction $T^{(k,j)}$ leads to a sequence of $F^{(k,j)}$ steps. Since the task is constant within the $j$-th episode, the episode's data distribution simplifies to $P_{\mathcal{S}\times\mathcal{A}}^{(k,j)}$. *Level 2 (Inter-episode)*: Across episodes, different tasks $T^{(k,j)}$ follow a distribution $P_T^{(k)}$. Thus, the overall data distribution on client $k$ is defined as:

$$P_{\mathcal{T}\times\mathcal{S}\times\mathcal{A}}^{(k)} = \sum\nolimits_{j=1}^{E^{(k)}} P_{\mathcal{S}\times\mathcal{A}}^{(k,j)} \cdot P_T^{(k)}\big(T^{(k,j)}\big), \qquad (9)$$

where $E^{(k)}$ is the number of episodes on client $k$. This two-level distribution captures richer, hierarchical patterns and introduces more severe skew than the one-level heterogeneity in traditional federated learning.

**Simplified Focus: Episode Length.** To study the above mentioned new heterogeneity in a tractable way, we simplify $P_T^{(k)}$ to reflect only the distribution of episode lengths $F^{(k,j)}$. That is, we consider how many steps each episode contains, rather than the task content itself. Ignoring this episode length heterogeneity can lead to misleading assumptions and subsequent degraded performance. For example, two clients might each have 10 episodes of shopping-related tasks. However, if one client's episodes are short and concise while the other's are long and repetitive, their training data contribute differently to the global model. This results in biased updates despite seemingly equal numbers of episodes. Moreover, even if step-length distributions are balanced, clients may differ in total episode count or task diversity, still causing skewed contributions.

To address this, we propose an adapted aggregation strategy in Section 3.3 that explicitly accounts for heterogeneity in episode step length, going beyond standard sample-count-based methods in traditional federated learning.

# D ADDITIONAL EXPERIMENTS & RESULTS

## D.1 EFFICIENCY EVALUATION ACROSS INFERENCE BACKENDS

**Setups.** To further investigate whether the annotation cost using our method can be reduced and whether the memory requirements can be minimized with current efficient inference backends, such as vLLM (Kwon et al., 2023) and LMDeploy (Contributors, 2023), we conduct additional experiments to assess efficiency by computing model costs and memory usage on different backends. API-based costs are assessed using the OpenAI's library tiktoken [1] to count input and output tokens via Auto-Annotation. The price per million tokens is also included. Moreover, we approximate the API cost for the Qwen2-VL family by using the pricing of Qwen-VL-Plus, as the server does not provide APIs for Qwen2-VL-7B-Instruct or Qwen2-VL-2B-Instruct. For the InternVL2 family, since the model server offers free trial access, we denote the cost as "Free". Note: the annotation costs are computed

---
[1] https://github.com/openai/tiktoken

| Annotation Model | Annotation Cost (¢) | | | Generation Time (s) | | Memory Usage (MB) | |
| --- | --- | --- | --- | --- | --- | --- | --- |
| | PyTorch | vLLM or LMDeploy | API | PyTorch | vLLM or LMDeploy | PyTorch | vLLM or LMDeploy |
| Human | 10880 | | | 56300 | | - | |
| GPT-4o-Mini | - | - | 14.8 | 5061 | | - | |
| GPT-4o | - | - | 247.92 | 6858 | | - | |
| Qwen2-VL-2B-Instruct | 6.14 | 8.42 | <21.15 | 1577 | 1180 | 12046 | 22083 |
| Qwen2-VL-7B-Instruct | 16.77 | 9.87 | <21.15 | 2005 | 1374 | 25881 | 22224 |
| InternVL2-1B | 2.58 | 11.09 | Free | 2000 | 1662 | 3985 | 20645 |
| InternVL2-2B | 16.04 | 7.23 | Free | 1698 | 1038 | 29235 | 21548 |
| InternVL2-8B | 23.18 | - | Free | 2245 | - | 31960 | - |

Table 6: Comparison of annotation costs per 1,000 samples, using different inference backends across various base models. Results demonstrate that employing efficient backends, such as vLLM and LMDeploy, can further reduce inference time and memory usage, ultimately lowering the annotation cost of our approach. The generation time and memory usage are averaged over three runs.

using Auto-Annotation-S, which removes the step-wise process for fair comparison across models. For reference, using vLLM, *Auto-Annotation* incurs around 2 times the cost of *Auto-Annotation-S*.

**Results.** As shown in Table 6, models exhibit explicitly different behaviors across backends. In general, most models reduce costs when using efficient backends. For example, InternVL2-2B saves annotation costs by more than half when leveraging LMDeploy. However, for smaller models, using an efficient backend does not necessarily lead to improvements. We attribute this to the fact that running vLLM on an RTX 4090 causes the model to occupy the entire GPU memory, which is 5 to 10 times the original memory usage of PyTorch. This increase in memory consumption does bring out improvement inference speed but fails to offset the additional memory demand. Since our annotation cost, as formulated in Equation 4.1, considers both time and memory usage, the overall cost does not necessarily decrease. Additionally, APIs remain a viable option since they eliminate the need for local deployment, while offering highly competitive pricing. However, using APIs comes at the sacrifice of privacy as shown in Table 1.

## D.2 CONTINUAL ABLATION EXPERIMENTS ON VARIOUS MODELS AND DATA SIZES

**Setups.** We further present our experiments, following Section 4.5. We conduct an ablation experiment on training data size to investigate whether the scaling law (Kaplan et al., 2020) holds for our automatically generated data. Using the AndroidControl dataset, we train models that differ only in the size of their training data. For MobileA3gent, we fix the number of clients at 10 and test different participation rates, specifically 30% and 90%. We also provide more comprehensive ablation on different model combinations in Figure 9. Both high-level and low-level settings are evaluated with *Type*, *Ground* and *SR* metrics. Details about our model suite are provided in Section F.4.

**Results.** As shown in Figure 8, the performance of all tested models improves as the training data size increases, indicating that our generated data also follows the scaling law. We also observe a sharp performance increase when training from 100 and 1,000 episodes. No saturation is observed in our experiments; however it can be inferred that the performance of all models grows more slowly once the data size reaches a certain threshold. Moreover, when comparing high-level and low-level training settings, the latter converges faster, due to its simplicity and less room for improvement

From Figure 9, (1) we further demonstrate that *Auto-Annotation* is an effective method for annotating user instructions. The generated data exhibits strong utility and can be scaled up significantly at minimal cost compared to manual labeling. (2) Increasing the data scale benefits the *Ground* metric the most, as it captures the most critical aspect that VLMs need to learn from training data—the grounding ability. Specifically, *Auto-Annotation* achieves up to an 82.8% improvement over *Human-Annotation* for InternVL2-1B.

## D.3 AUTO-ANNOTATION WITH DIFFERENT DATA SIZES

We present detailed experiments comparing *Auto-Annotation* with various baselines under two distinct data sizes. *Human-Annotation* serves as the upper bound.

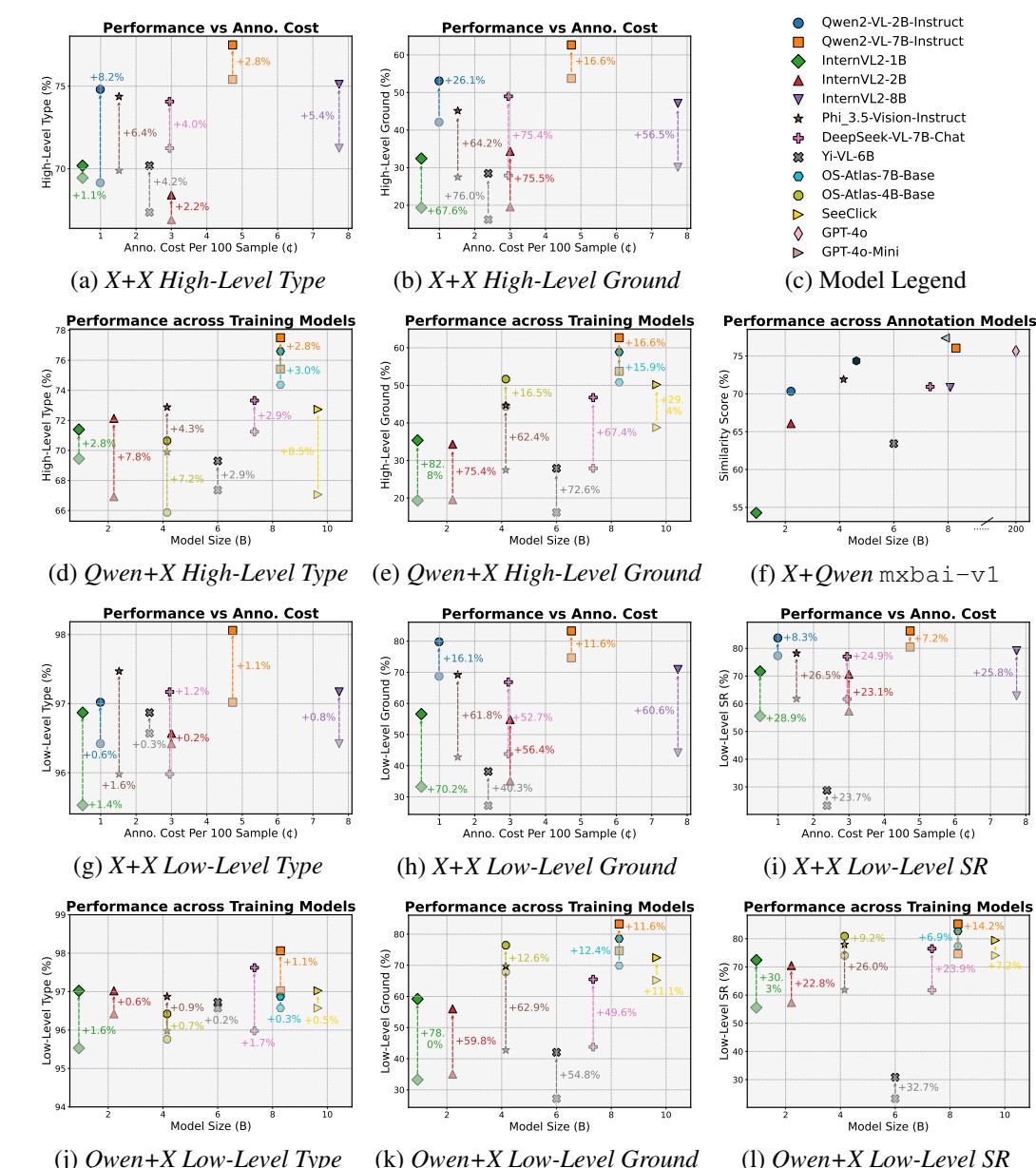

Figure 9: Comprehensive visualization of different base models evaluated across multiple metrics and training configurations.

**Comparison with Human-Annotation.** When the training data size is equal, our method achieves comparable performance on many evaluation metrics, with less than a 2% drop—for example, *SR* on both AndroidControl-High and AndroidControl-Low—while reducing annotation costs by over 99%. Moreover, as the data size scales up, our method surpasses *Human-Annotation* with ease, while still maintaining minimal cost.

**Comparison with Other Baselines.** *Auto-Annotation* maintain consistent superiority other baselines across all metrics and data sizes, making it the most effective choice for annotating user instructions.

### D.4 AUTO-ANNOTATION ON AITW DATASET

**Setups.** The AitW dataset consists of five subsets: General, Install, GoogleApps, Single, and WebShopping. For each subset of AitW, we sample 1,000 episodes for training and 100 for evaluation. The overall performance is the average of the five subsets. For the validation metric, we omit

| Methodology | AndroidControl-High | | | AndroidControl-Low | | | GUI Odyssey | | |
|---|---|---|---|---|---|---|---|---|---|
| | Type | Ground | SR | Type | Ground | SR | Type | Ground | SR |
| Qwen2-VL-7B-Instruct | 28.61 | 0.00 | 1.94 | 71.09 | 2.19 | 6.41 | 2.87 | 2.94 | 0.51 |
| GPT-4o (OpenAI, 2023) | 66.17 | 3.38 | 16.69 | 87.03 | 6.06 | 31.15 | 37.50 | 14.17 | 5.36 |
| OS-Atlas-7B-Pro (Wu et al., 2024) | 57.44 | 54.90 | 29.83 | 73.00 | 73.37 | 50.94 | 60.42 | 39.74 | 26.96 |
| | Data Size = 5,000 | | | | | | Data Size = 3,000 | | |
| Human-Annotation (Li et al., 2024a) | 79.14 | 66.56 | 61.70 | 97.62 | 81.47 | 85.99 | 84.39 | 75.63 | 67.01 |
| Action-Origin | 65.28 | 3.18 | 9.84 | 94.19 | 3.56 | 26.68 | 62.36 | 13.32 | 14.33 |
| Visual-Sense (Zhang et al., 2024a) | 77.49 | 61.13 | 57.37 | 97.47 | 81.42 | 85.54 | 81.53 | 67.66 | 59.49 |
| Self-Instruct (Wang et al., 2023) | 75.86 | 57.28 | 53.95 | 97.47 | 81.97 | 85.25 | 82.80 | 60.27 | 55.16 |
| Chain-of-Thought (Berkovitch et al., 2024) | 77.94 | 56.96 | 55.89 | 97.17 | 83.20 | 85.39 | 74.37 | 50.80 | 49.33 |
| Auto-Annotation | 77.49 | 62.67 | 58.12 | 98.06 | 83.29 | 86.29 | 81.72 | 69.51 | 60.57 |
| | Data Size = 1,000 | | | | | | | | |
| Human-Annotation (Li et al., 2024a) | 75.41 | 53.75 | 50.97 | 97.02 | 74.66 | 80.48 | 78.85 | 64.92 | 55.22 |
| Action-Origin | 65.28 | 2.85 | 10.58 | 90.61 | 1.14 | 28.46 | 54.52 | 5.38 | 7.52 |
| Visual-Sense (Zhang et al., 2024a) | 73.62 | 51.14 | 48.58 | 96.42 | 74.25 | 80.18 | 76.62 | 54.43 | 46.50 |
| Self-Instruct (Wang et al., 2023) | 72.43 | 48.99 | 47.54 | 96.87 | 72.40 | 78.69 | 77.07 | 51.33 | 45.22 |
| Chain-of-Thought (Berkovitch et al., 2024) | 72.58 | 48.48 | 47.24 | 97.02 | 74.53 | 80.18 | 76.56 | 53.40 | 46.37 |
| Auto-Annotation | 74.22 | 52.44 | 49.48 | 97.47 | 75.13 | 80.48 | 77.58 | 59.74 | 50.64 |

Table 7: In-depth evaluation of Auto-Annotation under equal data size on AndroidControl. In this setup, Human-Annotation serves as the upper bound due to its access to gold instructions. Auto-Annotation outperforms other baselines trained on model-annotated data and achieves comparable performance to Human-Annotation on several metrics-such as high-level *SR*-with drastic cost saving.

| Methodology | Size | General | Install | GoogleApps | Single | WebShopping | Overall |
|---|---|---|---|---|---|---|---|
| Zero-Shot | - | 15.90 | 5.20 | 15.08 | 28.38 | 11.41 | 15.19 |
| Human-Annotation | 1000 | 35.04 | 54.50 | 46.65 | 55.46 | 39.82 | 46.29 |
| Auto-Annotation-S | 1000 | 36.24 | 52.47 | 44.13 | 53.41 | 40.34 | 45.32 |
| Auto-Annotation | 1000 | 36.92 | 53.23 | 37.43 | 52.84 | 39.65 | 44.01 |
| Auto-Annotation-S | 5000 | 36.24 | 59.19 | 47.21 | 62.45 | 39.43 | 48.90 |
| Auto-Annotation | 5000 | 37.26 | 57.29 | 47.49 | 72.05 | 45.14 | 51.85 |

Table 8: Evaluation of Auto-Annotation across different subsets of AitW dataset. Our methods achieve consistent superior performance compared to Human-Annotation at a very low cost. -S denotes a simplified version which removes the step-wise description.

validation samples that consist of click actions with no corresponding unit. These samples are too easy to predict in our setting and show no meaningful difference between different models. We only present high-level accuracy due to the absence of low-level instructions in the original dataset.

**Results.** As shown in Table 8, we can conclude the following: (1) Apart from AndroidControl and GUI Odyssey, our method still achieves comparable results with Human-Annotation and outperform it by a large margin as the data size increases. (2) Our method performs extremely well on the *Single* subset. We attribute this result to the short average step length for episodes in *Single*, which leads to more accurate reconstruction of the high-level instructions.

## D.5 OUT-OF-DOMAIN EVALUATION WITH GENERALIZATION ANALYSIS

**Setups.** To evaluate the performance of MobileA3gent in out-of-domain scenarios, we conduct two experiments on the AndroidControl and GUI Odyssey datasets. For AndroidControl, we randomly sample 100 episodes from each of the three unseen test splits: *App-Unseen*, *Task-Unseen*, and *Category-Unseen*, based on the dataset's sub-splits. The number 100 is chosen to match the test sample size used in Section 4.3. For GUI Odyssey, we similarly sample 100 episodes from each unseen test split: *App-Unseen*, *Task-Unseen*, and *Device-Unseen*. Note that the original GUI Odyssey datasets include overlapping samples across splits; therefore, we select test episodes that do not overlap with either the training samples or with each other.

**Results.** As shown in Tables 9 and 10, (1) mobile agents trained on our automatically generated data exhibit strong generalizability across various settings. The results demonstrate the effectiveness of our approach and further validate the utility of our auto-annotated data, which is derived solely from screenshots and actions. (2) Additionally, we observe that the *Category-Unseen* subset yields

| Methodology | App-Unseen | | | Task-Unseen | | | Category-Unseen | | |
|---|---|---|---|---|---|---|---|---|---|
| | Type | Ground | SR | Type | Ground | SR | Type | Ground | SR |
| Human-Annotation (Li et al., 2024a) | 65.79 | 57.02 | 47.74 | 78.65 | 67.83 | 60.98 | 70.31 | 51.07 | 47.03 |
| Action-Origin | 56.48 | 5.92 | 9.90 | 64.21 | 0.99 | 12.75 | 60.16 | 1.88 | 7.81 |
| Visual-Sense (Zhang et al., 2024a) | 70.45 | 64.14 | 54.59 | 82.64 | **69.76** | 64.98 | 69.69 | **61.54** | 54.38 |
| Self-Instruct (Wang et al., 2023) | **72.63** | 64.06 | 56.04 | **84.18** | 67.12 | 64.06 | 69.84 | 59.45 | 53.75 |
| Chain-of-Thought (Berkovitch et al., 2024) | 71.03 | 58.33 | 51.97 | 82.64 | 68.42 | 64.21 | 71.09 | 59.22 | 55.47 |
| Auto-Annotation | 72.20 | **65.00** | 56.04 | 83.72 | 68.97 | **65.13** | **72.97** | 61.06 | **56.25** |

Table 9: Out-of-domain evaluation on AndroidControl. We compare Auto-Annotation with baselines on three out-of-domain test subsplits. Our method achieve consistent improvement over Human-Annotation with minimal annotation cost.

| Methodology | App-Unseen | | | Task-Unseen | | | Device-Unseen | | |
|---|---|---|---|---|---|---|---|---|---|
| | Type | Ground | SR | Type | Ground | SR | Type | Ground | SR |
| Human-Annotation (Li et al., 2024a) | **78.76** | 59.23 | 51.85 | 76.74 | 61.89 | 49.29 | 79.96 | 61.42 | 53.02 |
| Action-Origin | 61.54 | 9.94 | 11.35 | 62.56 | 11.22 | 11.63 | 61.84 | 11.86 | 12.52 |
| Visual-Sense (Zhang et al., 2024a) | 77.49 | 54.40 | 48.47 | 75.32 | 60.64 | 47.88 | 81.50 | 63.36 | 55.98 |
| Self-Instruct (Wang et al., 2023) | 78.00 | 49.33 | 45.22 | 76.96 | 56.75 | 46.58 | 82.24 | 57.72 | 52.40 |
| Chain-of-Thought (Berkovitch et al., 2024) | 77.36 | 53.81 | 48.21 | 77.24 | 56.71 | 46.53 | 82.31 | 57.98 | 53.08 |
| Auto-Annotation | 77.87 | **63.03** | **53.76** | **77.64** | **65.76** | **52.68** | **82.49** | **67.56** | **58.82** |

Table 10: Out-of-domain evaluation on GUI Odyssey. We compare Auto-Annotation with baselines on four evaluation subsets. Our method achieve consistent improvement over Human-Annotation with minimal annotation cost.

relatively lower accuracy compared to other evaluation subsets, indicating a higher level of difficulty. (3) For GUI Odyssey, we note that *Human-Annotation* achieves relatively higher performance than in other experiments, suggesting that this dataset may pose greater challenges for generalization.

## D.6 ACCURACY COMPARISON ACROSS DIFFERENT ACTIONS

Following the evaluation protocol described in Section 4.1, we compute the accuracy for each action type defined in the AndroidControl action space, as detailed in Table 11.

As illustrated in Figure 10, the accuracy varies significantly across different action types. Notably, the COMPLETE, TYPE, and OPEN_APP actions achieve relatively high accuracy. This can be attributed to the fact that these actions primarily depend on language understanding rather than visual grounding. Given that current VLMs are more proficient in handling language-based tasks, these actions are easier to infer correctly. In contrast, NAVIGATE_BACK and WAIT exhibit the lowest accuracy. We hypothesize that this is mainly due to their limited representation in the training set, as they constitute only a small portion of the total training data. Additionally, NAVIGATE_BACK often requires the model to correct previous errors or perform implicit reasoning based on prior steps, which is challenging for VLMs lacking explicit reasoning capabilities.

It is also worth noting that the *Type* metric differs fundamentally from the *SR* metric. The *Type* metric only requires correctly identifying the action type, without evaluating parameters such as coordinates or input content. In contrast, *SR* considers an action correct only if all its arguments are predicted accurately. This distinction is especially significant for coordinate-based actions like CLICK, which require precise location predictions to be considered successful under the *SR* metric. This additional complexity makes it more challenging for models to achieve high accuracy on such actions.

## E DISCUSSIONS AND FUTURE DIRECTIONS

### E.1 DISCUSSIONS

**Analysis of Resources on a Mobile Device.** To investigate the minimal resource requirements, we conduct additional experiments to determine whether small VLMs or models based on APIs can achieve similar effectiveness. The results in Section 4.5 show that even an 1B VLM can deliver competitive performance. Models based on APIs can also be used, though they come with the risk of privacy leakage, which we leave for future work.

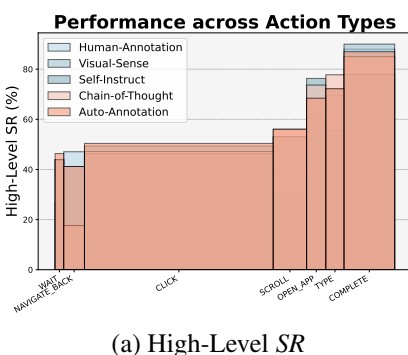 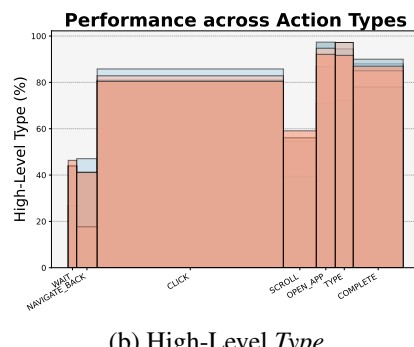

|                    |                    |
| ------------------ | ------------------ |
| (a) High-Level *SR* | (b) High-Level *Type* |

Figure 10: High-level *SR* across different action types within the action space of AndroidControl. The width of the pillars corresponds to the number of data samples in the evaluation test set; thus, the area reflects the weighted average performance.

**Real-World Applicability Analysis.** We will address the real-world applicability three-folds. First, as shown in Section 4.4, each user only needs to provide a small amount of data, and not all of it is sensitive, resulting in minimal or no privacy risks. Second, the benefits far outweigh the costs and risks. We assume that the server incentivizes participation by offering free use of the global agent in exchange for access to user data. Users can gain access to a highly capable mobile agent that saves them both time and efforts. Finally, by incorporating federated learning, user data is processed locally, alleviating most privacy concerns.

## E.2 FUTURE DIRECTIONS

As mentioned above, we have shown the promising results achieved by MobileA3gent. However, this is not the end as there are still emerging challenges and interesting directions that are worth exploring in this direction.

**Privacy Preservation.** Training on user data inevitably raises privacy concerns. While federated learning helps mitigate privacy leakage by keeping private data on the client side and transmitting only LoRA adapters, potential privacy issues remain. Models with substantial sizes are prone to memorization of their training data (Yu et al., 2024; Wang et al., 2024e). Similar to large LLMs, recent studies (Caldarella et al., 2024; Jayaraman et al., 2024) reveal that VLMs also inadvertently memorize and potentially expose sensitive information. Dejavu memorization (Jayaraman et al., 2024) proposes a novel measurement for memorization by quantifying the fraction of ground-truth objects in an image that can be predicted from its text description in a training image-text pair. Mobile agents rely on VLMs to perceive the interface and make decisions. Therefore, training directly on user data may lead to leakage of sensitive information. This issue can be addressed by implementing differential privacy (DP), which, however, remains underexplored in the context of VLMs and mobile agent training.

**Efficiency.** To collaboratively train a global mobile agent on distributed user data, each user needs to locally train a small-sized VLM and communicate with the central server. However, limited computation resources and communication channels on mobile devices may hinder the feasibility of deployment. With the recent advancement of LLMs and diffusion models and their integration into federated learning systems (Zhou et al., 2021), numerous approaches have been proposed to alleviate computational and communication overheads (Ding & Hu, 2024). On the other hand, the proliferation of smaller VLMs has significantly enhanced efficiency. For instance, AppVLM (Papoudakis et al., 2025) specifically targets app control tasks with a lightweight architecture, facilitating rapid and cost-efficient inference for real-time execution.

**Reinforcement Learning.** Although our current framework does not yet incorporate reinforcement learning, we identify it as a promising future direction. In a federated mobile agent setting, user feedback can serve as a critical reward signal, enabling agents to adjust their decision-making policies dynamically. Future work will need to tackle challenges inherent to integrating reinforcement learning into a federated environment, such as handling heterogeneous feedback, ensuring robust and stable

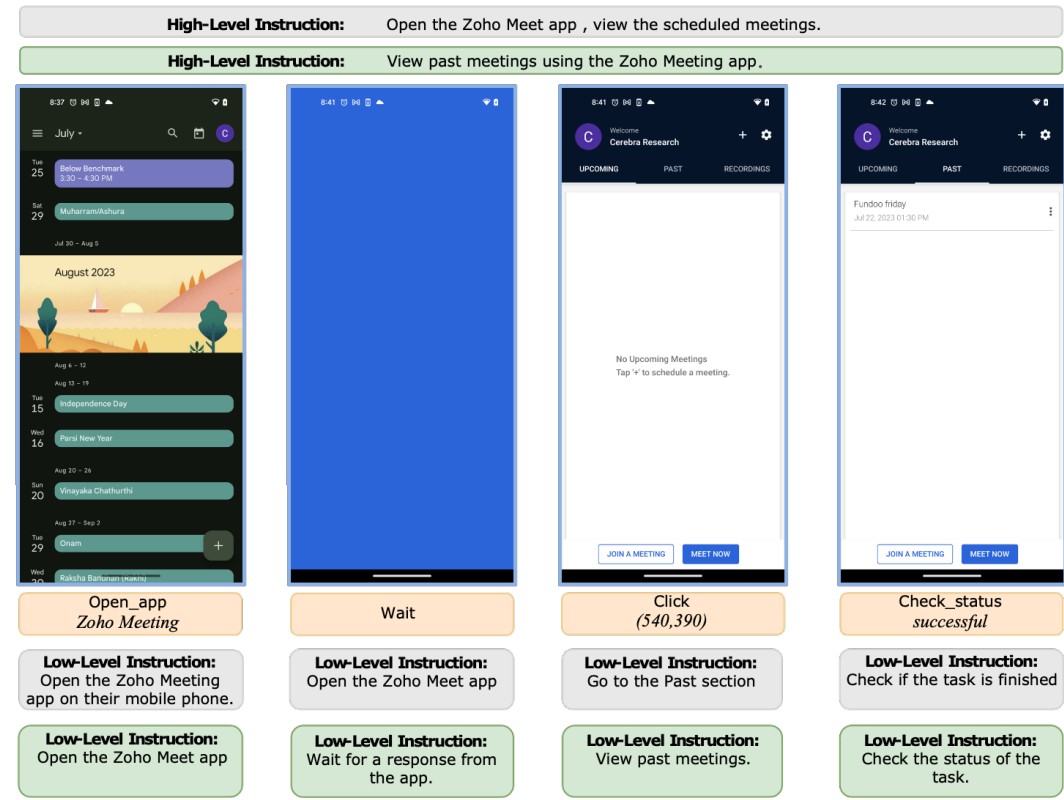

Figure 11: Episode example from Android Control dataset. The high-level task is *"Open the Zoho Meet app and view the scheduled meetings"*. Instructions in grey indicate ground truth from the original dataset, while those in green are predictions generated by Auto-Annotation. Our generated data sample achieves quality comparable to human-annotated ground truth.

learning under variable network conditions, and preserving user privacy. We believe that exploring these issues will pave the way for more adaptive and user-centric mobile agents, ultimately enhancing both their responsiveness and overall utility.

## F  EXPERIMENTAL DETAILS

### F.1  BENCHMARK DETAILS

To provide a comprehensive evaluation, we select four widely used mobile agent benchmarks from prior works (Wu et al., 2024; Sun et al., 2024; Zhang et al., 2024d), covering both offline and online settings.

**Offline Benchmarks.** In offline benchmarks, agents are evaluated using static screenshots and instructions under a step-wise evaluation protocol in a fixed order. Notably, even if an agent fails at a prior step that would normally prevent it from reaching the current step, the current step is still included in the evaluation. Offline benchmarks are favored in the GUI agent community due to their ease of quantification and deployment. We employ three widely accepted benchmarks from Google[2] and OpenGVLab[3].

- **AndroidControl (AC)** (Li et al., 2024a), evaluates agents' planning and action-execution capabilities in mobile environments. This benchmark provides two task types: (1) high-level tasks, where the agent must autonomously plan and execute multi-step actions; and (2) low-level tasks, where the agent is required to execute pre-defined, human-annotated actions which is more specific, at each step. During low-level tasks, both a low-level instruction and its corresponding high-level

---

[2]https://github.com/google-research/google-research
[3]https://github.com/OpenGVLab

instruction are included. We conduct experiments in both settings for a comprehensive assessment. A data example is provided in Figure 11 to further clarify the difference between high-level and low-level instructions.

- **Android in the Wild (AitW)** (Rawles et al., 2023), is a large-scale dataset annotated with instructional operations and screenshot-based icon detection, including element-level annotations generated using a pretrained IconNet. The AitW dataset comprises five subsets: General, Install, GoogleApps, Single, and WebShopping.
- **GUI Odyssey** (Lu et al., 2024b), focuses on cross-app navigation tasks in mobile environments, featuring an average of over 15 steps per task, which is notably longer than in AndroidControl. The tasks cover diverse navigation scenarios, and within each scenario, multiple instructions are generated based on predefined templates.

**Online Benchmark.** In contrast to offline benchmarks, online benchmarks prioritize realism and practical applicability. Agents are required to perform dynamic, interactive tasks in online simulation environments. And they continue attempting the task until reaching a predefined maximum step length. This setup may lead to some back-and-forth or repetitive behaviors as agents explore and recover from errors.

- **AndroidWorld (AW)** (Rawles et al., 2024), is an online environment designed for developing and benchmarking autonomous agents using a Pixel 6 phone simulator as the testbed. It comprises 116 tasks spanning 20 mobile apps, with dynamic task variations generated through randomized parameters. This dataset is particularly well-suited for evaluating agents' adaptability and planning abilities on mobile devices.

Our experimental setups for the offline datasets follow those in Wu et al. (2024), while the setups for the online benchmark adhere to the original implementation.

## F.2 Data Details

**Data Composition.** To offer a clearer understanding of the structure of mobile training datasets and the composition of a data episode, we present a representative example in Figure 11. As shown, each episode consists of: (1) A high-level instruction, expressed as a natural language sentence describing the task to be accomplished; (2) A sequence of low-level instructions, detailing the fine-grained tasks required for the current screenshot; notably, such annotations are only available in the AndroidControl dataset; (3) A series of screenshots captured from the start to the end of the task; and (4) A corresponding list of actions, aligned with the number of screenshots, indicating what the user does to progress to the next screenshot. All actions belong to an action space containing 7-9 options.

**Action Space.** Considering the action space used in OS-Atlas and the original AndroidControl paper, we define nine action types for AndroidControl. Notably, two of these action types, `Navigate_Home` and `Long_Press`, appear only rarely. For GUI Odyssey, one more action type `Press_Recent` is defined as press the recent button to switch between different apps as most tasks are cross-app. For the AitW dataset, we define seven action types. The corresponding actions and their descriptions are provided in Tables 11 and 13, with any additional parameters indicated as *Target*. In AitW, we decompose the original `Press` action into three distinct actions: `Navigate_Home`, `Navigate_Back`, and `Press_Enter`, aligning the action space with that of AndroidControl. Additionally, we derive the `Scroll` action from the original dual-point action.

**Splits.** Regarding training and testing splits, for AndroidControl, we adopt the original splits provided in the paper[4]. Specifically, we sample 5,000 episodes for training and 100 episodes for each test subsplit, i.e., *IID*, *App-Unseen*, *Task-Unseen*, and *Category-Unseen*. Unless otherwise specified, our results (except for the generalization experiments reported in Section D.5) are evaluated based on the *IID* subsplit. For each subset of AitW, we sample 1,000 episodes for training and 100 for evaluation.

## F.3 Metrics Details

**Offline Metrics.** To facilitate fair comparisons across all baseline methods, we standardize the evaluation metrics for all action types. For each step, we provide three metrics: *Type*, *Ground* and

---

[4]`https://console.cloud.google.com/storage/browser/gresearch/android_control`

| Action Type | Attribute | Description |
|---|---|---|
| **Basic Actions** | | |
| CLICK | (x,y) | Click at a specific point on the screen using the coordinates. |
| TYPE | text | Type the text in the current input field or search bar. |
| SCROLL | direction | Scroll in a specific direction (one of 'up', 'down', 'left', or 'right'). |
| **Custom Actions** | | |
| LONG_PRESS | (x,y) | Long press at a specific point on the screen using the coordinates. |
| NAVIGATE_BACK | - | Return to the previous page or undo an action. |
| NAVIGATE_HOME | - | Return to the home page. |
| OPEN_APP | app_name | Open an app with the specified name. |
| WAIT | - | Pause for a moment before proceeding with the next action. |
| COMPLETE | - | Indicate that the task is finished. |

Table 11: Action space for AndroidControl.

| Action Type | Attribute | Description |
|---|---|---|
| **Basic Actions** | | |
| CLICK | (x,y) | Click at a specific point on the screen using the coordinates. |
| TYPE | text | Type the text in the current input field or search bar. |
| SCROLL | direction | Scroll in a specific direction (one of 'up', 'down', 'left', or 'right'). |
| **Custom Actions** | | |
| LONG_PRESS | (x,y) | Long press at a specific point on the screen using the coordinates. |
| NAVIGATE_BACK | - | Return to the previous page or undo an action. |
| NAVIGATE_HOME | - | Return to the home page. |
| PRESS_RECENT | - | Press 'Recent' to switch between recently used applications. |
| WAIT | - | Pause for a moment before proceeding with the next action. |
| COMPLETE | - | Indicate that the task is finished. |

Table 12: Action space for GUI Odyssey.

*SR*. Continual on the description in Section 4.1, we further detail on how an action is determined as correct for *SR*.

- For coordinate-related actions, e.g. `Click`, the agents generate both the action type and the position coordinates. Since the ground-truth bounding box is not always available, we measure the performance by computing the distance between the predicted coordinates and the ground-truth coordinates. Following Bai et al. (2024), we deem the coordinates correct if they fall within a distance equivalent to 14% screen width from the ground truth.
- For type-based actions (e.g., `TYPE`, `OPEN_APP`), we compute the F1 score between the predicted text and the ground truth. A prediction is considered correct if the F1 score exceeds $0.5$.
- For `SCROLL` actions, the direction argument (i.e., `UP`, `DOWN`, `LEFT`, or `RIGHT`) must precisely match the ground truth.
- For all other actions (e.g., `PRESS_BACK`), the prediction must exactly match the ground truth to be considered correct.

**Online Metrics.** The evaluation is conducted in screenshot-only mode. To mitigate potential interference from network instability and environmental factors, the results are measured three times. The primary metric is the episode-wise task success rate, a more rigorous measurement compared to the step-wise success rate (*SR*) in offline mode, as en episode is considered successful only when all constituent steps are performed correctly, i.e. *SR* = 100% for a task to be successful.

**Data Quality Metrics.** Based on the well established literature in NLP community. We use similarity of generated instruction to the ground truth as an indication of data quality. We adopt both text-based metrics which directly computed based on the two sentences and embedding-based metrics.

- **BLEU** (Bilingual Evaluation Understudy) is a precision-based metric that evaluates text similarity by comparing n-grams between generated and reference texts (Papineni et al., 2002).
- **ROUGE** (Recall-Oriented Understudy for Gisting Evaluation) is a recall-based metric that computes overlapping n-grams, word sequences, and the longest common subsequences (Lin,

| Action Type | Attribute | Description |
|---|---|---|
| **Basic Actions** | | |
| CLICK | (x,y) | Click at a specific point on the screen using the coordinates. |
| TYPE | text | Type the text in the current input field or search bar. |
| SCROLL | direction | Scroll in a specific direction (one of 'up', 'down', 'left', or 'right'). |
| **Custom Actions** | | |
| NAVIGATE_BACK | - | Return to the previous page or undo an action. |
| NAVIGATE_HOME | - | Return to the home page. |
| PRESS_ENTER | - | Press the 'Enter' button. |
| COMPLETE | - | Indicate that the task is finished. |
| IMPOSSIBLE | - | Indicate that the task is infeasible. |

Table 13: Action space for Android in the Wild.

2004). The ROUGE family includes ROUGE-1, ROUGE-2, and ROUGE-L, each providing measures for precision, recall, and the F1-score.

- **TF-IDF** (Term Frequency-Inverse Document Frequency) is a statistical measure that evaluates word importance in a document relative to a corpus by balancing term frequency and inverse document frequency (Salton & Buckley, 1988).
- **METEOR** (Metric for Evaluation of Translation with Explicit ORdering) is a metric that evaluates text similarity by aligning unigrams between generated and reference texts using exact, stem, synonym, and paraphrase matches. Unlike BLEU, METEOR incorporates both precision and recall, along with a fragmentation penalty to account for word order, resulting in higher correlation with human judgments at the sentence level (Banerjee & Lavie, 2005).
- **Embedding Similarity** which use embedding models to embed the sentences first and calculates the cosine similarity between two embedding vectors. We select two SOTA embedding models with the most downloads on the Hugging Face websites, `jina-v3`[5] and `mxbai-v1`[6].

## F.4    MODEL DETAILS

We employ three categories of models in our experiments: VLMs with conversational capability, base models specialized for GUI tasks with enhanced grounding ability, and API-based closed-ended models.

- **Chat Models.** We select widely used VLMs from prior and contemporary works (Bai et al., 2024; Sun et al., 2024). Specifically, we include the Qwen2-VL family (2B[7], 7B[8]) (Wang et al., 2024c), InternVL2 family (1B[9], 2B[10], 4B[11], 8B[12]) (Chen et al., 2024b), DeepSeek-VL-7B-Chat[13] (Lu et al., 2024a), Phi-3.5-Vision-Instruct[14] from Microsoft (Abdin et al., 2024), Ovis2-4B[15] from AIDC-AI (Lu et al., 2024c), and Yi-VL-6B[16], an early model from 01-AI.
- **GUI Base Models.** We adopt SeeClick[17] (Cheng et al., 2024), which is continually pre-trained on Qwen-VL-7B with additional grounding datasets from ScreenSpot (Cheng et al., 2024). We also utilize OS-Atlas-4B[18] and OS-Atlas-7B[19] (Wu et al., 2024), which are trained on InternVL2-4B

---

[5]https://huggingface.co/jinaai/jina-embeddings-v3
[6]https://huggingface.co/mixedbread-ai/mxbai-embed-large-v1
[7]https://huggingface.co/Qwen/Qwen2-VL-2B-Instruct
[8]https://huggingface.co/Qwen/Qwen2-VL-2B-Instruct
[9]https://huggingface.co/OpenGVLab/InternVL2-1B
[10]https://huggingface.co/OpenGVLab/InternVL2-2B
[11]https://huggingface.co/OpenGVLab/InternVL2-4B
[12]https://huggingface.co/OpenGVLab/InternVL2-8B
[13]https://huggingface.co/deepseek-ai/deepseek-vl-7b-chat
[14]https://huggingface.co/microsoft/Phi-3.5-vision-instruct
[15]https://huggingface.co/AIDC-AI/Ovis2-4B
[16]https://huggingface.co/01-ai/Yi-VL-6B
[17]https://huggingface.co/cckevinn/SeeClick
[18]https://huggingface.co/OS-Copilot/OS-Atlas-Base-4B
[19]https://huggingface.co/OS-Copilot/OS-Atlas-Base-7B

and Qwen2-VL-7B-Instruct, respectively. These models lack conversational capabilities and are therefore unsuitable for annotation.

- **API-Based Models.** GPT-4o and GPT-4o-Mini (OpenAI, 2023) are widely used vision models provided by OpenAI. These models are significantly more cost-effective than GPT-4V and are frequently utilized in researches. Due to their closed-source and API-only nature, they do not support supervised fine-tuning within our framework and are exclusively used as annotation models.

### F.5 BASELINE DETAILS

**Overall Baselines for Training Mobile GUI Agents.** In Section 4.2, we compare existing approaches for data collection and mobile agent training. In this section, we provide further elaboration and details on these baselines.

- **Human-Annotated Data.** Most conventional approaches fall into this category, which involves first employing crowdsourcing to collect and annotate data, followed by training mobile GUI agents. Depending on the training paradigm—centralized or federated—this category can be further divided into two baselines: *Central-Human* and *FedLLM/VLM*. To the best of our knowledge, no prior work has explored training federated VLMs. Therefore, we extend the existing FedLLM framework to the FedVLM setting while retaining the name *FedLLM* for consistency and comparison.
- **Synthetic Data.** This approach (Sun et al., 2024; Su et al., 2025) leverages VLMs to generate synthetic instructions, either based on seed task-driven instructions annotated by humans or through reverse task synthesis. These synthetic instructions are subsequently executed in simulators, by either powerful models such as GPT-4o or by humans, to collect full interaction trajectories. *OS-Genesis* (Sun et al., 2024) is a representative example of this category. Although these methods substantially reduce human labor, they still heavily rely on powerful API-based models and extensive simulator execution, which can become costly at scale.
  Due to the unavailability of the original training data, we are unable to directly evaluate *OS-Genesis* within our setting. Instead, we reference reported results from the original paper. For cost estimation, we measure the cost of generating a single data sample using GPT-4o in our setup and extrapolate it to 1,000 samples (the dataset size used in OS-Genesis), yielding the $\approx 10^3$ cost estimates presented in Table 2.
- **DistRL*.** *DistRL* (Wang et al., 2024d) proposes a scalable and asynchronous architecture for data acquisition from multiple simulators in a distributed manner, coupled with centralized reinforced agent training. The framework also introduces techniques to compensate for potential performance degradation caused by asynchrony. We adapt this method to our user-based setting by collecting auto-annotated data in a distributed manner using the *Auto-Annotation* mechanism and training the model centrally. We refer to this adapted baseline as *DistRL*. The key distinction between *MobileA3gent* and *DistRL** lies in the training paradigm, and the latter raises greater privacy concerns due to the exposure of user data to both peers and the server during centralized training.

**Annotation Baselines.** We compare five baselines for annotating user instructions based on available information, including screenshots and action sequences.

- **Action-Origin,** directly concatenates the original formatted actions into a text string without any inference, representing the simplest method for retrieving user instructions in natural language.
- **Visual-Sense,** (Zhang et al., 2024a) leverages the visual perception capabilities of the annotation VLM to understand the screenshots recorded during task execution. Specifically, we concatenate the sequence of screenshots into one image and feed it into the annotation model for one-shot inference.
- **Self-Instruct,** (Wang et al., 2023) is originally proposed for synthetic data generation using LLMs. We adapt it to infer user intentions from action sequences. In our implementation, all actions are provided simultaneously to the annotation model, which predicts the instruction in a single pass.
- **Chain-of-Thought,** (Berkovitch et al., 2024) guides the annotation model (e.g., GPT-4o) through a step-by-step reasoning process to analyze the task trajectory. At each step, the model predicts the current intention based on all prior information, and the final instruction is determined after the entire task sequence is completed. It is important to note that, although named "Chain-of-Thought," this method is derived from (Berkovitch et al., 2024), which focuses on identifying user intentions in GUI tasks, rather than from the original CoT prompting paper (Wei et al., 2022).

| Parameter | Value | Parameter | Value |
|---|---|---|---|
| **Federated Learning** | | | |
| number-of-rounds | 30 | number-of-clients | 10 |
| number-of-clients-sampled | 3 | ratio $\lambda$ | 3,5,7,9 |
| **LoRA Configuration** | | | |
| lora-rank | 8 | lora-alpha | 32 |
| lora-dropout | 0.05 | max-sequence-length | 4096, 2048 |
| **Optimization** | | | |
| learning-rate | $5 \times 10^{-5}$ | batch-size | 1 |
| optimizer | adamw_torch | gradient-accumulation-steps | 4 |
| weight-decay | 0.1 | adam-beta1 | 0.9 |
| adam-beta2 | 0.95 | adam-epsilon | $1 \times 10^{-8}$ |
| lr-scheduler | cosine | warmup-ratio | 0.03 |
| **Quantization Settings** | | | |
| bnb-4bit-compute-dtype | torch.bfloat16 | bnb-4bit-quant-type | nf4 |
| bnb-4bit-use-double-quant | true | load-in-4bit | false |
| load-in-8bit | false | device-number | 2 |

Table 14: Key training parameters regarding FL, LoRA, and quantization.

- **Human-Annotation,** uses human-annotated gold instructions from the dataset, serving as the upper-bound reference. However, with increasing data scale, methods based on automatic annotation, including ours, can not only achieve comparable or even superior performance, but also substantially reduce annotation costs.

**Federated Learning Baselines.** We integrate seven representative federated learning algorithms, following the implementations provided in OpenFedLLM (Ye et al., 2024). These include FedAvg (McMahan et al., 2017), FedProx (Li et al., 2020), SCAFFOLD (Karimireddy et al., 2020), FedAvgM (Hsu et al., 2019), FedAdagrad, FedYogi, and FedAdam (Reddi et al., 2020).

- **Local Update.** FedAvg is the foundational algorithm upon which many subsequent methods are built. FedProx and SCAFFOLD extend FedAvg by incorporating local model correction mechanisms to mitigate the effects of data heterogeneity.
- **Global Aggregation.** In contrast, FedAvgM, FedAdagrad, FedYogi, and FedAdam introduce server-side momentum techniques to stabilize global model updates.
- **Local Training.** Additionally, we include a local training baseline, where a model is trained solely on a single client's dataset without collaboration. This serves as a reference to highlight the benefits of participating in federated learning.

### F.6 TRAINING AND GENERATION DETAILS

**Training Setups.** The models are trained over 10 rounds, with each round processing one-tenth of the total dataset. This setup ensures that, in expectation, each data sample is seen approximately once throughout the training process.

In the IID federated learning setting, data samples are uniformly distributed across 10 or more clients. In each round, 30% of clients are randomly selected to perform local training and participate in global aggregation. Analogous to centralized training, each selected client processes one-tenth of its local data during that round. Therefore, training for 10 rounds yields an expected 30% overall client participation. To simulate higher participation (e.g., 90%), we extend training to 30 rounds. While in non-IID setting (e.g., experiments in Section 4.4), the data samples are distributed according to the specific scenario.

For experiments investigating the effect of dataset size and scaling, we start with an initial pool of 5,000 data samples. Subsets of smaller sizes are created by selecting the first $X$ samples from this pool to form datasets of size $X$. This approach guarantees that datasets with larger sample sizes always encompass those with fewer samples, ensuring consistency and comparability across experiments.

**Training Framework.** We build upon the highly-starred training framework, ms-swift (Zhao et al., 2024) [20], and extend it into a repository capable of training federated VLMs. Our extension follows the implementation of federated training framework for Large Language Models (LLMs) (Ye et al., 2024). We apply Low-Rank Adaptation (LoRA) (Hu et al., 2021) to improve efficiency.

**Training Parameters.** As shown in Table 14, we include all key parameters for reproducibility. For max-sequence-length, we choose 4096 for Qwen2-VL family and 2048 for InternVL2 family. The hyperparameter for various federated algorithms are set as: FedYogi (Reddi et al., 2020) employs momentum factors ($\beta_1 = 0.9, \beta_2 = 0.999$) with learning rate $\eta = 10^{-3}$ and stabilization constant $\tau = 10^{-6}$. FedAvgM (Hsu et al., 2019) uses $0.9/0.1$ ratio for historical/current model interpolation. FedProx (Li et al., 2020) applies proximal regularization with $\mu = 0.2$ through $||w - w^t||^2$ penalty terms. SCAFFOLD (Karimireddy et al., 2020) configurations maintain server learning rate $\eta_s = 1.0$ with client momentum compensation, while FedAdam and FedAdagrad (Reddi et al., 2020) share base parameters ($\beta_1 = 0.9, \beta_2 = 0.999$) with adaptive learning rate scaling. All algorithms expose tunable coefficients through the framework's unified parameter interface.

**Templates.** We provide all of our prompt templates used in generating instructions and training. Specifically, generation prompts for *Auto-Annotation* are in Figures 12, 13; generation prompt for *Visual-Sense* is provided in Figure 14 with *Chain-of-Thought* in Figure 15; training prompts are shown in Figure 16, 17 and 18 for all three offline datasets respectively.

---

[20]https://github.com/modelscope/ms-swift

**Prompt 1: Step-Wise Description**

A user is performing a *task* on a mobile phone, progressing through **multiple steps** to complete the task.
Each step involves an interface shown in the provided screenshot, and an action performed to move on to the next step.

Based on the screenshot and the user's action, infer the specific goal the user is trying to accomplish at this step in the task.
You need to associate the action with the key information in the screenshot and output your predicted goal.

## Example
- User Action: Scroll down
if the screenshot shows the browsing page for purchasing shoes,
- Your Output: Swipe up for more product details about shoes

- User Action: Click (101,314)
if the UI element at this coordinate is an article titled "cooking"
- Your Output: Click on the article titled "cooking"

- User Action: Check status: successful
- Your Output: Check if the task is finished

- User Action: Open App: Plantum
if the action is *open app*, return the same
- Your Output: Open App: Plantum

- User Action: Wait for response
if the action is *wait*, return none
- Your Output: None

## Answer Format
Only output the predicted goal. Be specific with the input screenshot.
Keep your response concise and capture the important things, focusing on key details like the app name, email address, search terms, item name, and title.

## User Action
{converted action $A_i$ }

## Your Output
The user is trying to:

Figure 12: Prompt template for the Descriptor to generate low-level instruction $\mathcal{T}_i^{low}$ based on the converted action $A_i$ and screenshot $s_i$ at the $i$-th step .

---

**Prompt 2: Episode-Wise Summarization within Auto-Annotation**

A user is performing a high-level **task** on a mobile phone, progressing through multiple low-level steps to complete the task.
Each step involves an interface, and a low-level action performed to move on to the next step.

The full sequence of user actions is provided in the *History* section.
The **task** is not known. Now based on the history provided, describe the mobile user's high-level **task** when performing these actions.

## History
{ low-level instruction $\mathcal{T}_1^{low}$ }
{ low-level instruction $\mathcal{T}_2^{low}$ }
...
{ low-level instruction $\mathcal{T}_n^{low}$ }

## Answer Format
Keep your output concise and clear, as if the user were explaining the **task** to someone else in one sentence.
Include key details like the app name, individual name, email address, search terms, item name, and title.

## Your Output
The user is trying to:

---

Figure 13: Prompt template for the Summarizer to generate high-level instruction $\mathcal{T}^{high}$ based on the list of low-level instructions and the concatenated screenshot $s_c$.

---

**Prompt 3: Episode-Wise Summarization with Visual-Sense**

A user is performing a high-level **task** on a mobile phone, progressing through multiple low-level steps to complete the task.
Each step involves an interface, and a low-level action performed to move on to the next step.

A single image that shows all the screenshots concatenated horizontally is provided.
The **task** is not known. Now based on this concatenated screenshot, describe the mobile user's high-level **task** when performing these actions.

## Answer Format
Keep your output concise and clear, as if the user were explaining the **task** to someone else in one sentence.
Include key details like the app name, individual name, email address, search terms, item name, and title.

## Your Output
The user is trying to:

---

Figure 14: Prompt template for *Visual-Sense* to generate high-level instruction $\mathcal{T}^{high}$ based on the list of converted actions and the concatenated screenshot $s_c$.

**Prompt 4: Step-Wise Description with Chain-of-Thought**

A user is performing a high-level **task** on a mobile phone, progressing through multiple low-level steps to complete the task.
Each step involves an interface, and a low-level action performed to move on to the next step.

The previous task descriptions for each step are provided in the *History* section, and the user's final action is provided in the *User Action* section.
You need to think step by step and analyze the input sequence to deduce the user's underlying objective that prompted these actions.
Utilize the screenshot of the final step to gain insights into the user's intentions, focusing on elements highlighted or implicated by the actions.
Your goal is to describe the ultimate intention the user is aiming to achieve.

## History
{ low-level instruction $\mathcal{T}_1^{low}$ }
{ low-level instruction $\mathcal{T}_2^{low}$ }
...
{ low-level instruction $\mathcal{T}_{i-1}^{low}$ }

## User Action
{converted action $A_i$ }

## Answer Format
Keep your output concise and clear, as if the user were explaining the **task** to someone else in one sentence.
Include key details like the app name, individual name, email address, search terms, item name, and title.

## Your Output
The user is trying to:

Figure 15: Prompt template for *Chain-of-Thought* to generate instruction step-by-step and finally obtain the high-level instruction.

**Prompt 5: Common Prompt for Training**

You are a foundational action model capable of automating tasks across various digital environments, including desktop systems like Windows, macOS, and Linux, as well as mobile platforms such as Android and iOS. You also excel in web browser environments. You will interact with digital devices in a human-like manner: by reading screenshots, analyzing them, and taking appropriate actions.

Your expertise covers two types of digital tasks:

- Grounding: Given a screenshot and a description, you assist users in locating elements mentioned. Sometimes, you must infer which elements best fit the description when they aren't explicitly stated.

- Executable Language Grounding: With a screenshot and task instruction, your goal is to determine the executable actions needed to complete the task.

You are now operating in Executable Language Grounding mode. Your goal is to help users accomplish tasks by suggesting executable actions that best fit their needs. Your skill set includes both basic and custom actions:

1. Basic Actions
Basic actions are standardized and available across all platforms. They provide essential functionality and are defined with a specific format, ensuring consistency and reliability.

- Basic Action 1: CLICK
    - purpose: Click at the specified position.
    - format: `CLICK <point>[[x-axis, y-axis]]</point>`
    - example usage: `CLICK <point>[[101, 872]]</point>`
- Basic Action 2: TYPE
    - purpose: Enter specified text at the designated location.
    - format: `TYPE [input text]`
    - example usage: `TYPE [Shanghai shopping mall]`
- Basic Action 3: SCROLL
    - purpose: Scroll in the specified direction.
    - format: `SCROLL [direction (UP/DOWN/LEFT/RIGHT)]`
    - example usage: `SCROLL [UP]`

Figure 16: Prompt template for the common part shared between different datasets during training of federated mobile agents within MobileA3gent. The full training prompt is the combination of the common part and the custom part.

**Prompt 6: Custom Prompt for Training on AndroidControl**

2. Custom Actions
Custom actions are unique to each user´s platform and environment. They allow for flexibility and adaptability, enabling the model to support new and unseen actions defined by users. These actions extend the functionality of the basic set, making the model more versatile and capable of handling specific tasks.

- Custom Action 1: LONG_PRESS
    - purpose: Long press at the specified position.
    - format:                    `LONG_PRESS <point>[[x-axis, y-axis]]</point>`
    - example usage:             `LONG_PRESS <point>[[272, 341]]</point>`
- Custom Action 2: NAVIGATE_BACK
    - purpose: Press a back button to navigate to the previous screen.
    - format: `NAVIGATE_BACK`
    - example usage: `NAVIGATE_BACK`
- Custom Action 3: NAVIGATE_HOME
    - purpose: Press a home button to navigate to the home page.
    - format: `NAVIGATE_HOME`
    - example usage: `NAVIGATE_HOME`
- Custom Action 4: OPEN_APP
    - purpose: Open the specified application.
    - format: `OPEN_APP [app_name]`
    - example usage: `OPEN_APP [Google Chrome]`
- Custom Action 5: WAIT
    - purpose: Wait for the screen to load.
    - format: `WAIT`
    - example usage: `WAIT`
- Custom Action 6: COMPLETE
    - purpose: Indicate the task is finished.
    - format: `COMPLETE`
    - example usage: `COMPLETE`

In most cases, task instructions are high-level and abstract. Carefully read the instruction and action history, then perform reasoning to determine the most appropriate next action. Ensure you strictly generate two sections: **Thoughts** and **Actions**.

**Thoughts**: Clearly outline your reasoning process for current step.
**Actions**: Specify the actual actions you will take based on your reasoning.

Your current task instruction, action history, and associated screenshot are as follows:
Screenshot: <image>
Task: {high-level instruction $\mathcal{T}^{high}$}
You need to: {low-level instruction $\mathcal{T}_i^{low}$}
History: {history of $\mathcal{T}_i^{low}$}

Figure 17: Custom prompt template for training mobile GUI agents on AndroidControl.

---

**Prompt 7: Custom Prompt for Training on GUI Odyssey**

Custom actions are unique to each user's platform and environment. They allow for flexibility and adaptability, enabling the model to support new and unseen actions defined by users. These actions extend the functionality of the basic set, making the model more versatile and capable of handling specific tasks.

- Custom Action 1: LONG_PRESS
  - purpose: Long press at the specified position.
  - format: `LONG_PRESS <point>[[x-axis, y-axis]]</point>`
  - example usage: `LONG_PRESS <point>[[272, 341]]</point>`
- Custom Action 2: NAVIGATE_BACK
  - purpose: Press a back button to navigate to the previous screen.
  - format: `NAVIGATE_BACK`
  - example usage: `NAVIGATE_BACK`
- Custom Action 3: NAVIGATE_HOME
  - purpose: Press a home button to navigate to the home page.
  - format: `NAVIGATE_HOME`
  - example usage: `NAVIGATE_HOME`
- Custom Action 4: PRESS_RECENT
  - purpose: Press the recent button to view or switch between recently used applications.
  - format: `PRESS_RECENT`
  - example usage: `PRESS_RECENT`
- Custom Action 5: WAIT
  - purpose: Wait for the screen to load.
  - format: `WAIT`
  - example usage: `WAIT`
- Custom Action 6: COMPLETE
  - purpose: Indicate the task is finished.
  - format: `COMPLETE`
  - example usage: `COMPLETE`

In most cases, task instructions are high-level and abstract. Carefully read the instruction and action history, then perform reasoning to determine the most appropriate next action. Ensure you strictly generate one section: **Actions**.

**Actions**: Specify the actual actions you will take based on your reasoning.
Your current task instruction, action history, and associated screenshot are as follows:
Screenshot: <image>
Task: {high-level instruction $\mathcal{T}^{high}$}

---

Figure 18: Custom prompt template for training mobile GUI agents on GUI Odyssey.

