# OpenReview forum: "MobileA3gent: Training Mobile GUI Agents Using Decentralized Self-Sourced Data from Diverse Users"
_ICLR.cc/2026/Conference — Submitted to ICLR 2026_

### Official Review · Reviewer_QgZJ · 2025-10-28

**Soundness:** 2
**Presentation:** 3
**Contribution:** 3
**Rating:** 6
**Confidence:** 3

**Summary:**

This paper presents a federated learning framework for collaboratively training mobile GUI agents using clients’ self-sourced data. It proposes an auto-annotation mechanism to enable the automatic collection of training datasets during users' routine phone usage, and a federated VLM training mechanism to enable privacy-preserving collaborative agent training on distributed user data.

**Strengths:**

1.	**Well-organized and clearly written:** The paper is well-structured and easy to follow, with clear motivation and methodology presentation.
2.	**Practical and meaningful research problems:** The work focuses on automatic data annotation and collaborative agent evolution—two significant and practically relevant challenges for building intelligent mobile systems.
3.	**Extensive experiments:** The paper provides relatively comprehensive experiments.

**Weaknesses:**

1.	**Practical feasibility concerns:** Although the proposed framework is conceptually appealing, it may face challenges in real-world deployment. Performing both annotation and training on mobile devices could incur high computational and energy overhead. It is unclear whether such resource consumption is acceptable on mobile devices, nor its potential impact on other on-device applications’ responsiveness.
2.	**Methodology:** The proposed approach is relatively simple and appears more inspirational than technically rigorous. Key technical aspects are not deeply explored.
3.	**Noisy annotations:** It is unclear how the framework handles low-quality training samples generated by the proposed auto-annotation mechanism to ensure the robustness and stability of the learned model.
4.	**Limited real-world evaluation:** The evaluation setup relies on manually partitioned datasets to simulate mobile devices, which limits its realism. Moreover, some experiments are conducted under IID settings, making the scenario even less representative of real-world federated environments. The client scale is also relatively small.

**Questions:**

1.	How feasible is it for mobile devices to handle both auto-annotation and local training in terms of computational cost, battery consumption, and latency?
2.	Could the authors provide more realistic or large-scale experiments to better reflect real-world mobile federated scenarios?

---

> ### Author Response · Authors · 2025-11-22
> **Response to Reviewer QgZJ (1/4)**
>
> Thank you for recognizing our paper and for the time and effort you devoted to reviewing it. With your valuable feedback and expertise, we believe we can further mitigate the limitations of traditional approaches and move toward a more cost-effective and scalable solution.
> Below, we address each of your concerns accordingly.
>
> ---
> &nbsp;
>
> ## Response to Weakness 1 and Question 1: Practical Concern & Resources
>
> > W1: Although the proposed framework is conceptually appealing, it may face challenges in real-world deployment.
> > It is unclear whether such resource consumption is acceptable on mobile devices, nor its potential impact on other on-device applications’ responsiveness.
>
> > Q1: How feasible is it for mobile devices to handle both auto-annotation and local training in terms of computational cost, battery consumption, and latency?
>
> Thank you for raising this point.
> We address this concern by elaborating on the feasibility of real-world deployment from two perspectives:
>
> &nbsp;
>
> ### Time and Resource Statistics for Annotation and Training
>
> We provide detailed resource statistics, including memory usage and generation time (latency), in **Table 4, Appendix D.1**.
>
> For your convenience, we reproduce the results in Table A. The measurements are based on 1,000 samples. For users with only 50 trajectories, both time and cost are significantly reduced, **taking roughly one minute**.
>
> [**Table A**. Time and resource usage for both training and annotation.]
> |Model|Generation GPU (MB)|Generation Time/1000 Epi. (s)|Train GPU (MB)| Train Time/1000 Epi. (hours)|
> |--|--|--|--|--|
> |Phi-3.5-Vision-Instruct|9686|1374|11558|5.21|
> |InternVL2-1B|3985|1662|10330|3.15|
> |Qwen2-VL-2B-Instruct|12046|1180|15680|2.58|
>
> Notably, Intern2-VL-1B has the lowest GPU requirements, needing only 4GB for generation and 10GB for training. This makes it **feasible for many mobile devices with 16GB of RAM or more**.
> Larger models, such as Phi-3.5-Vision-Instruct and Qwen2-VL-2B-Instruct, can also **fit within a 16 GB capacity**.
> Moreover, these requirements can be further reduced by adopting efficient training backbones.
>
> &nbsp;
>
> ### On-Device Deployability Supported by Substantial Research
>
> 1. Importantly, according to many technical reports (e.g. Phi models [1], MobileAgent-V3 [2] and AppCopilot [3]), the relative small VLM models including *Phi-3.5 and InternVL2-1/2B*, are actually **deployable on mobile devices**.
> 2. Moreover, an increasing number of studies have focused on developing lightweight models tailored for mobile tasks, such as LiMAC [4] and AppVLM [5]. These efforts toward building smaller, more efficient VLMs further support the practical viability of on-device GUI agents.
>
> &nbsp;
>
> ### Summary
>
> In conclusion, the deployment of VLMs on mobile devices is increasingly feasible, driven by advancements in model efficiency and growing support for lightweight architectures.
> Therefore, based on our detailed resource statistics, we believe the resource consumption is acceptable for mobile devices, and that it is highly feasible for them to support both auto-annotation and local training.
>
> &nbsp;
>
> ### Reference
>
> [1] Phi-3 Technical Report: A Highly Capable Language Model Locally on Your Phone. Microsoft, 2024.
> [2] Mobile-Agent-v3: Fundamental Agents for GUI Automation, 2025.
> [3] AppCopilot: Toward General, Accurate, Long-Horizon, and Efficient Mobile Agent, 2025.
> [4] Lightweight Neural App Control. ICLR 2025.
> [5] AppVLM: A Lightweight Vision Language Model for Online App Control. 2025.
>
> ---

---

> ### Author Response · Authors · 2025-11-22
> **Response to Reviewer QgZJ (2/4)**
>
> ## Response to Weakness 2: Methodology
>
> > The proposed approach is relatively simple and appears more inspirational than technically rigorous. Key technical aspects are not deeply explored.
>
> We apologize for any potential misunderstanding.
> The proposed **user-centric approach is indeed inpired** by the practical insights to utilize everday phone usage, but the **MobileA3gent framework is rigorously designed and validated** through systematic analyses and experiments.
> Our contributions span two main components: Auto-Annotation and FedVLM-A.
>
> 1. **Auto-Annotation**
>
> For Auto-Annotation, we are the first to introduce a hierarchical generation pipeline that produces human-like instructions from user phone trajectories. As demonstrated in our experiments, this annotation strategy leads to consistently improved performance over methods like CoT and Self-Instruct.
>
> 2. **FedVLM-A**
>
> For FedVLM-A, we first articulate the new forms of heterogeneity introduced in the setting of federated mobile agents, **Two-Level Distribution**, with detailed formulations provided in **Appendix C.2 and C.3**.
> Building upon this analysis, we further propose a principled extension of FedAvg that preserves its convergence properties while achieving better adaptation to the newly identified heterogeneity.
>
> We believe the overall architecture of our system is both novel and coherent. While more sophisticated designs may further improve performance, as the first work to formalize the problem of federated mobile agents, we contend that our current design already demonstrates strong empirical promise. We hope our efforts can inspire subsequent research and foster deeper exploration in this emerging direction.
>
> ---
> &nbsp;
>
> ## Response to Weakness 3: Annotation Noise
>
> > It is unclear how the framework handles low-quality training samples generated by the proposed auto-annotation mechanism to ensure the robustness and stability of the learned model.
>
> &nbsp;
>
> ### Denoising Technique Implementation
>
> Following your valuable advice, we have implemented new techniques to handle potential noise samples that could negatively impact the VLM during annotation.
>
> Our implementation effectively tackles the most common and impactful noise sources—stuck states and repetitive navigation patterns—which are prevalent across diverse mobile interaction scenarios.
> **Please refer to our response to Reviewer JzkA, Weakness 3** for implementation details.
>
> While we recognize that there are numerous sources of noise in real-world interaction data and that our current implementation may not address every possible noise pattern, our systematic approach to trajectory cleaning has demonstrated significant practical benefits.
>
> &nbsp;
>
> ### Clarification of the Primary Problem and Future Directions
>
> Also we would like to note that, the primary problem of this paper is *how to harness distributed user trajectory with minimal human intervention*.
> Since MobileA3gent is the initial work in the direction of user-centric GUI agent, our priority is to establish a foundation with normalized data before addressing the challenges associated with noisy data.
> We need to first answer the question, *How can we make this possible?* before delving into *How can we handle more noise types?*.
> We sincerely hope you may consider our major contribution, which includes both the proposal of the task and the complete architecture we have implemented.
>
> &nbsp;
>
> We hope that the techniques we have added to handle low-quality training samples adequately address your concern.
>
> ---

---

> ### Author Response · Authors · 2025-11-22
> **Response to Reviewer QgZJ (3/4)**
>
> ## Response to Weakness 4 and Question 2: Realistic and Large-Scale Experiments
>
> &nbsp;
>
> ### More Realistic, Complex and Large-Scale Experiment
>
> > W4: The evaluation setup relies on manually partitioned datasets to simulate mobile devices, which limits its realism.
>
> > Q2: Could the authors provide more realistic or large-scale experiments to better reflect real-world mobile federated scenarios?
>
> Thank you for the insightful comments. We first note that our appendix already includes larger-scale experiments (more data samples) in **Table 5** and out-of-distribution experiments in **Table 7 & 8**.
> Following your suggestion, we further extend our evaluation by incorporating a more realistic and large-scale federated setting.
>
> To better approximate real-world mobile data distributions within the limits of publicly available datasets, we construct a combined federated environment by merging three existing mobile datasets. Specifically, we randomly partition **6,000 episodes with 59,328 steps** from Android Control, AitW, and GUI Odyssey into **100 clients**.
>
> These clients exhibit high heterogeneity in both **data characteristics** (originating from different apps/datasets) and **data scales** (varying numbers of steps and episodes).
>
> [**Table E**. Realistic experiments combining three datasets with 59328 steps across 100 clients.]
> |Methodology|AndroidControl-High|||AndroidControl-Low|||GUI Odyssey|||Avg.|||
> |--|--|--|--|--|--|--|--|--|--|--|--|--|
> ||Type|Ground|SR|Type|Ground|SR|Type|Ground|SR|Type|Ground|SR|
> | Central | 78.30 | 54.05 | 55.08 | 72.50 | 69.87 | 58.02 | 87.77 | 77.01 | 69.78 | 79.52 | 66.98 | 60.96 |
> | Local | 65.40 | 7.74 | 22.00 | 62.81 | 43.68 | 36.56 | 73.90 | 43.22 | 34.68 | 67.37 | 31.55 | 31.08 |
> | FedAvg | 73.60 | 31.93 | 42.19 | 72.08 | 55.03 | 52.19 | 81.73 | 64.20 | 55.29 | 75.80 | 50.39 | 49.89 |
> | Fedprox | 74.51 | 31.96 | 42.49 | 71.88 | 53.97 | 51.56 | 81.04 | 63.68 | 54.53 | 75.81 | 49.87 | 49.53 |
> | FedVLM-A | 74.51 | 34.59 | 43.55 | 72.40 | 56.59 | 52.85 | 81.59 | 64.54 | 55.56 | 76.17 | 51.91 | 50.65 |
>
>
> As shown in Table E:
> 1. Federated learning algorithms experience larger performance gaps compared to centralized learning due to the substantial number of clients and severe heterogeneity.
> 2. Nevertheless, **FedVLM-A consistently matches or surpasses other FL baselines**, and federated learning significantly outperforms local training.
>
> &nbsp;
>
> ### Non-IID Experiment
>
> > W4: Moreover, some experiments are conducted under IID settings, making the scenario even less representative of real-world federated environments.
>
> We explain that certain experiments were intentionally conducted under IID settings to control for distribution-induced variance, enabling fairer comparison of Auto-Annotation performance and validating the feasibility of federated mobile agents.
> Experiments in Figure 5, Section 4.4, by contrast, are already performed under non-IID conditions.
>
> To further address this concern, we additionally evaluate non-IID scenarios constructed from Android Control. Specifically, We categorize all episodes into five semantic groups (i.e. Shopping, Travelling, Office, Lives, and Entertainment), based on the operating app. We then create **three data-distribution settings** over five clients:
> - **IID**: Each client receives an equal number of episodes from every app category.
> - **Skew**: Each client observes only one unique category.
> - **Non-Uniform**: All clients observe all categories, but with highly imbalanced category proportions.
>
> [**Table F**. Non-IID experiments on Android Control, with data distributed by episode category.]
> | Heterogeneity Type | Algorithm | Shopping | Travelling | Office | Lives | Entertainment | Avg. |
> |---|---|---|---|---|---|---|---|
> |-|Zero-Shot|26.61|25.33|27.05|24.41|23.81|25.46|
> ||Central|57.26|58.67|51.64|55.12|60.95|56.90|
> |IID|Local|48.39|45.78|36.89|32.28|45.71|42.25|
> ||FedAvg|55.65|52.00|52.46|37.80|51.43|50.07|
> ||FedProx|53.23|52.44|51.64|38.58|51.43|49.79|
> ||FedVLM-A|54.84|52.89|50.00|38.58|49.52|49.64|
> |Skew|Local|50.81|47.56|46.72|38.58|48.57|46.51|
> ||FedAvg|52.42|52.00|48.36|41.73|52.38|49.64|
> ||FedProx|51.61|52.44|47.54|41.73|49.52|49.08|
> ||FedVLM-A|54.84|52.89|48.36|42.52|52.38|50.50|
> |Non-Uniform|Local|38.71|33.78|34.43|34.65|33.33|34.85|
> ||FedAvg|50.00|48.89|47.54|40.94|46.67|47.08|
> ||FedProx|47.94|52.42|50.22|45.90|42.52|46.67|
> ||FedVLM-A|50.39|51.56|48.72|43.31|48.57|48.51|
>
>
> From the results in Table F, we conclude that:
> 1. Category heterogeneity exists and degrades federated learning performance, as nearly all algorithms show a performance drop when transitioning from homogeneous
> to heterogeneous scenarios (from IID -> Skew).
> 2. **Across multiple heterogeneity patterns, beyond the two-level heterogeneity discussed in Section 4.4**, FedVLM-A consistently achieves competitive or superior performance relative to baselines such as FedAvg and FedProx.

---

> ### Author Response · Authors · 2025-11-22
> **Response to Reviewer QgZJ (4/4)**
>
> ## Response to Weakness 4 and Question 2: Realistic and Large-Scale Experiments
>
> &nbsp;
>
> ### Larger Scale Experiment
>
> > W4: The client scale is also relatively small.
>
> To adequately address your concern about larger scale experiments, we expand both the data volume and the number of clients in Table E.
> In addition, we conduct an ablation study **specifically targeting the impact of client size** to examine whether involving more user participation leads to improved federated performance in MobileA3gent.
>
> In this experiment, we vary the number of clients while keeping a fixed budget of 100 episodes per client. Thus, increasing the client count results in a larger total amount of training data.
>
> [**Table G**. Experiments with increasing client numbers.]
> | Client Number | 10    | 30    | 50    | 70    |
> |--------------|-------|-------|-------|-------|
> | Client Sample| 1     | 3     | 5     | 7     |
> | FedAvg       | 51.81 | 56.06 | 57.17 | 57.48 |
>
> The results in Table G demonstrate that as the number of clients increases, the overall model performance improves accordingly, further validating the scalability and effectiveness of the proposed framework.
>
> &nbsp;
>
> ### Out-of-Distribution Experiment
>
> To further assess generalization, we additionally report selected out-of-distribution (OOD) results in Table H for your reference. Experiment details are provided in Appendix D.5.
>
> [**Table H**. Out-of-distribution experiments on GUI Odyssey.]
> |Methodology| App-Unseen ||| Task-Unseen ||| Device-Unseen |||
> |-----|----|----|------|-----|-----|------|--------|------|------|
> | | Type | Ground | SR | Type | Ground | SR | Type | Ground | SR |
> | Human-Annotation | 78.76 | 59.23 | 51.85 | 76.74 | 61.89 | 49.29 | 79.96 | 61.42 | 53.02 |
> | Self-Instruct | 78.00 | 49.33 | 45.22 | 76.96 | 56.75 | 46.58 | 82.24 | 57.72 | 52.40 |
> | Auto-Annotation | 77.87 | 63.03 | 53.76 | 77.64 | 65.76 | 52.68 | 82.49 | 67.56 | 58.82 |
>
> As shown in Table H, mobile GUI agents trained on automatically generated data
> exhibit **strong generalizability across various settings**. The results validate the utility of our auto-annotated data and further demonstrate the effectiveness of MobileA3gent.
>
> ---
> &nbsp;
>
> We hope that our clarifications on practical feasibility, along with the additional realistic, large-scale, and non-IID experiments, effectively address your concerns. We sincerely appreciate your thoughtful feedback and the valuable insights you provided.
> If you have any additional suggestions, we would be more than pleased to adopt them and further improve this paper. We would be very grateful if you might consider increasing the score.

---

> ### Author Response · Authors · 2025-11-27
> **Kind Reminder**
>
> Dear Reviewer QgZJ,
>
> We gently invite you to review our additional experiments (enhanced in realism, scale, and heterogeneity) along with our clarifications.
> Your feedback is invaluable in strengthening our work. We hope these additions sufficiently address your concerns, and we would be happy to provide any further explanations if needed.
>
> We sincerely appreciate your time and consideration,
>
> The Authors

---

### Official Review · Reviewer_ybJy · 2025-10-30

**Soundness:** 3
**Presentation:** 3
**Contribution:** 3
**Rating:** 6
**Confidence:** 4

**Summary:**

This work proposes MobileA3gent, a framework for training mobile GUI agents from self-sourced usage data while preserving privacy. It has two main components:

(1) Auto-Annotation, which runs a local VLM to infer user intent from interaction trajectories using a step-wise Descriptor plus episode-wise Summarizer; and
(2) FedVLM-A, a federated training framework that adapts global aggregation by jointly weighting episode and step counts to better handle heterogeneous user data.

Experiments on multiple mobile-agent benchmarks show that MobileA3gent matches or surpasses human-labelled training at significantly lower cost, improves robustness under non-IID settings, and preserves privacy by keeping data on-device

**Strengths:**

* Practical FL innovation: Introduces an adapted aggregation method that jointly weights episode and step counts, effectively addressing two-level data heterogeneity with a simple, interpretable design.

* The system is technically sound and thoroughly validated across four benchmarks with non-IID splits, ablations, and scaling analyses showing consistent gains over FL baselines.

* Well-written and easy to follow, supported by clear figures and organized methodology.

* Achieves human-annotation-level accuracy at much lower labeling cost, demonstrating strong scalability and privacy preservation. The paper also discusses ethical risks transparently.

**Weaknesses:**

* No on-device evaluation for either annotation or training, leaving open questions on latency, energy, and memory feasibility.

* Annotation models (e.g., Qwen2-VL-7B) exceed mobile capacity, creating a gap between intended on-device use and evaluated setups.

**Questions:**

* What exactly is transmitted (gradients, LoRA deltas, or weights)? Any clipping, noise, or secure aggregation to preserve privacy?

* What are the latency and VRAM for Qwen2-VL-2B and Qwen2-VL-7B for annotation and training on edge hardware?

---

> ### Author Response · Authors · 2025-11-22
> **Response to Reviewer ybJy (1/2)**
>
> Thank you for recognizing our paper and for the time and effort you devoted to reviewing it. With your valuable feedback and expertise, we believe we can further mitigate the limitations of centralized approaches and move toward a more cost-effective and scalable solution.
> Below, we address each of your concerns accordingly.
>
> ---
> &nbsp;
>
> ## Response to Weakness 1: Efficiency Analysis
>
> > No on-device evaluation for either annotation or training, leaving open questions on latency, energy, and memory feasibility.
>
> Thank you for pointing out this important aspect.
> As acknowledged in our limitation section (Section B.2), it is of high difficulty for us to run real-device experiment due to the lack of avaiable experiment devices and current immaturity of the community’s training codebase.
> However we have already conducted detailed evaluation towards important efficency metrics based on the experiment on cloud servers, which have a similar demonstration of how the resources look like on mobile phones.
>
> &nbsp;
>
> ### Time and Resource Statistics for Annotation
>
> We provide detailed resource statistics, including memory usage and generation time (**latency**), in **Table 4, Appendix D.1**.
> For your convenience, we reproduce the table here. The measurements are based on 1,000 samples. For users with only 50 trajectories, both time and cost are significantly reduced, **taking roughly one minute**.
>
> As shown in Table D, **deploying InternVL2-1B requires only 3GB of memory**, making it feasible for mobile devices equipped with 8GB RAM or more.
>
> [**Table D**. Time and resource usage for annotation.]
> | Annotation Model   | Annotation Cost (¢)    |       |     | Generation Time (s)          |       | Memory Usage (MB)      |           |
> |--------|---------|----|-------|--------|---------------|-------------|-----------|
> |      | PyTorch    | vLLM or LMDeploy | API   | PyTorch                      | vLLM or LMDeploy | PyTorch                    | vLLM or LMDeploy |
> | Human     |         |        | 10880 | 56300                        |               |      |   |
> | GPT-4o-Mini             | -      | -             | 14.8  | 5061           |               | -                           | -             |
> | GPT-4o      | -            | -             | 247.92| 6858                         |               | -       | -             |
> | Qwen2-VL-2B-Instruct    | 6.14                             | 8.42          | <21.15| 1577  | 1180          | 12046                       | 22083         |
> | Qwen2-VL-7B-Instruct    | 16.77                            | 9.87          | <21.15| 2005      | 1374          | 25881                       | 22224         |
> | InternVL2-1B            | 2.58                             | 11.09         | Free  | 2000       | 1662          | 3985                        | 20645         |
> | InternVL2-2B            | 16.04                            | 7.23          | Free  | 1698  | 1038          | 29235                       | 21548         |
> | InternVL2-8B            | 23.18    | -             | Free  | 2245        | -             | 31960                       | -             |
>
> &nbsp;
>
> ### Time and Resource Statistics for Training
>
> We further analyze the training statistics in Table A (a shared table also included in our response to Reviewer QgZJ).
>
> [**Table A**. Time and resource usage for both training and annotation.]
> | Model                | Generation GPU (MB) | Generation Time/1000 Epi. (s) | Train GPU (MB) | Train Time/1000 Epi. (hours) |
> |----------------------|---------------|------------------------|---------------|---------------------------|
> | Phi-3.5-Vision-Instruct | 9686          | 1374                   | 11558         | 5.21                      |
> | InternVL2-1B         | 3985          | 1662                   | 10330         | 3.15       |
> | Qwen2-VL-2B-Instruct | 12046         | 1180                   | 15680         | 2.58 |
>
> Notably, Intern2-VL-1B has the lowest GPU requirements, needing only 4GB for generation and 10GB for training. This makes it **feasible for many mobile devices with 16GB of RAM or more**.
> Larger models, such as Phi-3.5-Vision-Instruct and Qwen2-VL-2B-Instruct, can also **fit within a 16 GB capacity**.
> Moreover, these requirements can be further reduced by adopting efficient training backbones.
>
> &nbsp;
>
> We hope that the above statistics satisfactorily answer your questions on latency, energy, and memory feasibility.
>
> ---

---

> ### Author Response · Authors · 2025-11-22
> **Response to Reviewer ybJy (2/2)**
>
> ## Response to Weakness 2: Using 7B Model
>
> > Annotation models (e.g., Qwen2-VL-7B) exceed mobile capacity, creating a gap between intended on-device use and evaluated setups.
>
> Sorry for the potential misunderstanding.
> We address this concern and elaborate on the feasibility of on-device model deployment from two perspectives:
>
> &nbsp;
>
> ### Model-Agnostic Design with Experiments on Diverse Models
>
> We indeed use Qwen2-VL-7B as the main model for our experiments, as it delivers strong performance and is widely adopted in prior GUI agent research. This choice allows us to more reliably validate the effectiveness of our proposed framework.
>
> However, we clarify that **MobileA3gent is model-agnostic**, and we also conduct extensive experiments with **a diverse set of VLMs of various sizes**, as shown in **Section 4.5 (Figure 6) and Appendix D.2 (Figure 9)**.
>
> Specifically, we evaluate models from the Qwen2-VL family (2B, 8B), InternVL2 family (1B, 2B, 4B, 8B), DeepSeek-VL-7B-Chat, Phi-3.5-Vision-Instruct (4B), Ovis2-4B, and Yi-VL-6B. For GUI-specialized base models, we further include SeeClick, as well as OS-Atlas-4B and OS-Atlas-7B.
> All model configurations and details are provided in **Appendix F.4**.
>
> &nbsp;
>
> ### On-Device Deployability Supported by Substantial Research
>
> 1. Importantly, according to many technical reports (e.g. Phi models [1], MobileAgent-V3 [2] and AppCopilot [3]), the relative small VLM models including *Phi-3.5 and InternVL2-1/2B*, are actually **deployable on mobile devices**.
> 2. Moreover, an increasing number of studies have focused on developing lightweight models tailored for mobile tasks, such as LiMAC [4] and AppVLM [5]. These efforts toward building smaller, more efficient VLMs further support the practical viability of on-device GUI agents.
>
> &nbsp;
>
> ### Summary
>
> In summary, while we adopt Qwen2-VL-7B for stronger empirical validation, our framework is model-agnostic and is thoroughly evaluated across a wide spectrum of VLM sizes, and existing research has already demonstrated the feasibility of deploying lightweight VLMs on mobile devices.
>
> We hope these clarifications collectively resolve your concern about using the 7B model.
>
> ---
> &nbsp;
>
> ## Response to Question 1:
>
> > What exactly is transmitted (gradients, LoRA deltas, or weights)?
>
> We transmit LoRA adapters/deltas for communication effiency.
> Please refer to our response to Reviewer JzkA, Weakness 4, for **additional statistics on communication overhead and frequency**.
>
> > Any clipping, noise, or secure aggregation to preserve privacy?
>
> In the initially submitted version, we did not employ these techniques. Following your suggestion, we have **implemented secure aggregation** as described in [6] by adding a pre-calculated random gradient before uploading the LoRA adapters. This mitigates gradient inversion attacks, as adversaries cannot access the client’s actual gradients.
> We will update this part in the privacy-related section (i.e. Section 3.3) and revise the architecture figure once all suggestions are addressed.
>
> Additionally, we have provided **a detailed privacy analysis**, including the threat model, adversary assumptions, and privacy protection mechanisms, in our response to Reviewer k9bV, Weakness 3. Please refer to that for more details.
>
> ---
> &nbsp;
>
> ## Response to Question 2:
>
> > What are the latency and VRAM for Qwen2-VL-2B and Qwen2-VL-7B for annotation and training on edge hardware?
>
> The annotation latency for 1,000 samples is 1,180 s for Qwen2-VL-2B and 1,374 s for Qwen2-VL-7B. The corresponding VRAM usage is 12,046 MB and 25,881 MB, respectively.  For training, VRAM requirements are 15,680 MB and 36,744 MB. These measurements were obtained using two RTX 4090 GPUs; using a single GPU for training would yield lower VRAM usage.
>
> Notably, **Qwen2-VL-2B, even without any efficient design optimizations, can already fit within the 16 GB VRAM limit of typical modern phones** (e.g., 16 GB RAM with 512 GB storage). For additional results, please refer to Tables A and B in our responses.
>
> We hope these results help address your questions.
>
> ---
> &nbsp;
>
>
> ## Reference
>
> [1] Phi-3 Technical Report: A Highly Capable Language Model Locally on Your Phone. Microsoft, 2024.
> [2] Mobile-Agent-v3: Fundamental Agents for GUI Automation, 2025.
> [3] AppCopilot: Toward General, Accurate, Long-Horizon, and Efficient Mobile Agent, 2025.
> [4] Lightweight Neural App Control. ICLR 2025.
> [5] AppVLM: A Lightweight Vision Language Model for Online App Control. 2025.
> [6] Practical Secure Aggregation for Privacy-Preserving Machine Learning, CCS 2017.
>
> ---
> &nbsp;
>
> Overall, we hope we have successfully addressed you concerns. If you have further suggestions, we are more than pleased to adopt them and improve this paper.
> We appreciate your feedback and will be grateful if you might consider increasing the score.

---

> ### Author Response · Authors · 2025-11-27
> **Kind Reminder**
>
> Dear Reviewer ybJy,
>
> Thank you for your constructive feedback on our paper. With the ICLR public discussion phase ending shortly, we wanted to confirm whether our responses have sufficiently addressed your concerns. If there are any remaining issues, we would be happy to provide additional clarifications.
>
> Thank you very much for your time and consideration!
>
> The Authors

---

### Official Review · Reviewer_JzkA · 2025-10-31

**Soundness:** 3
**Presentation:** 3
**Contribution:** 3
**Rating:** 4
**Confidence:** 3

**Summary:**

This paper introduces MobileA3gent, a framework for training mobile GUI agents by leveraging self-sourced data from users’ daily smartphone interactions in a privacy-preserving and cost-efficient manner.

1. Auto-Annotation – A local vision-language model (VLM)–based method that automatically infers user intentions (instructions) from GUI interaction trajectories, decomposing them into step-wise atomic descriptions and episode-level summaries to produce high-quality labeled data without human effort.

2. FedVLM-A – A federated learning framework that trains VLM-based mobile agents across distributed user devices using an adapted global aggregation scheme. This scheme accounts for both episode-level and step-level heterogeneity in user data distributions, improving over standard FL approaches like FedAvg or FedYogi.

Experiments across four benchmarks (AndroidControl, Android in the Wild, GUI Odyssey, AndroidWorld) and 10+ models show that MobileA3gent achieves comparable or superior performance to centralized, human-annotated baselines while reducing annotation costs by up to 99%. The system also provides strong privacy guarantees and scalability via decentralized participation.

**Strengths:**

The work addresses a pressing issue in the GUI agent community — the cost and scalability of data collection for training agents. By enabling distributed, self-sourced data annotation, the paper proposes a paradigm shift toward user-centric, privacy-preserving model training.

The Auto-Annotation mechanism is conceptually novel, integrating low-level (step-wise) and high-level (episode-wise) VLM reasoning to generate human-like task instructions.

The Adapted Aggregation in FedVLM-A introduces a meaningful extension to traditional FL weighting, explicitly handling two-level heterogeneity (episodes and steps) — a nontrivial characteristic of GUI data.

The experiments are extensive and rigorous, covering multiple datasets, model families, and evaluation metrics. Ablation studies demonstrate the impact of each component (Auto-Annotation, FedVLM-A) and the trade-offs between cost and performance.

MobileA3gent matches or exceeds centralized human-annotation baselines while being vastly more efficient. Results on AndroidWorld demonstrate solid generalization to unseen environments.

The paper is well-structured, with clear motivation, method description, and visualizations. The commitment to releasing code and the detailed appendices further support reproducibility.

**Weaknesses:**

Although the system is motivated by decentralized real-user data, all experiments use public datasets, not actual on-device data collection. Thus, claims about privacy, scalability, and real-world feasibility remain unverified empirically. The gap between simulation and deployment is significant.

While FedVLM-A preserves privacy by design, the paper provides no formal privacy analysis (e.g., differential privacy bounds or adversarial leakage evaluation). Table 1’s qualitative comparison is insufficient for a paper emphasizing privacy.

It remains ambiguous how annotation errors propagate through federated training. There is no analysis of annotation noise tolerance or mechanisms to correct systematically wrong inferences made by local VLMs.

Federated VLM training is resource-intensive. The paper omits a detailed analysis of client-side computation, communication frequency, or energy cost, which are critical for feasibility on mobile devices.

**Questions:**

this dataset curation method might contain sparse and noisy data, i wonder how does it compare to centralized manual data curation

---

> ### Author Response · Authors · 2025-11-22
> **Response top Reviewer JzkA (1/3)**
>
> Thank you for recognizing our paper and for the time and effort you devoted to reviewing it. With your valuable feedback and expertise, we believe we can further mitigate the limitations of centralized approaches and move toward a more cost-effective and scalable solution.
> Below, we address each of your concerns accordingly.
>
> ---
> &nbsp;
>
> ## Response to Weakness 1: On-Device User Data Collection
>
> > Although the system is motivated by decentralized real-user data, all experiments use public datasets, not actual on-device data collection.
>
> Thank you for highlighting this important aspect.
> As mentioned in our limitation section (Section B.2), collecting user data directly from devices presents significant challenges due to ethical concerns and logistical constraints.
>
> The main difficulty lies in *balancing the need for real user data to build more realistic and representative datasets with the need to protect user privacy*. Using real user data would indeed provide more accurate usage patterns that are valuable for academic research, but it also raises considerable privacy and ethical issues.
> In contrast, open-source datasets facilitate direct comparison with existing work and pose no barriers to public release, but may not fully capture the authenticity of
> real-world usage.
>
> Given the ethical concerns and the high costs associated with collecting user data, we opted for the latter approach: using publicly available datasets for our experiments. This choice **ensures that we can still advance the research while maintaining a responsible and ethical stance**.
>
> ---
> &nbsp;
>
> ## Response to Weakness 2: Privacy Analysis
>
> > While FedVLM-A preserves privacy by design, the paper provides no formal privacy analysis (e.g., differential privacy bounds or adversarial leakage evaluation).
>
> Thank you for raising this point. We recognize that the original submission could not provide a detailed discussion on privacy due to page limitations.
> To address the privacy concerns shared by you and Reviewer k9bV, we have added **a dedicated and more comprehensive privacy analysis**, covering:
> - the threat model and assumptions regarding potential adversaries,
> - the data that are stored or transmitted within the system, and
> - the corresponding privacy protection mechanisms.
>
> These additions clarify the specific privacy guarantees offered by our system and we hope they address the concern raised in the review.
> Please refer to **our response to Reviewer k9bV, Weakness 3** for further details.
>
> Additionally, we have implemented **secure aggregation** to further strengthen privacy protection against potential attacks targeting gradients. Please see **our response to Reviewer ybJy, Question 1** for more information.
>
> ---

---

> ### Author Response · Authors · 2025-11-22
> **Response top Reviewer JzkA (2/3)**
>
> ## Response to Weakness 3: Annotation Noise
>
> > There is no analysis of annotation noise tolerance or mechanisms to correct systematically wrong inferences made by local VLMs.
>
> &nbsp;
>
> ### Denoising Technique Implementation
>
> Following your valuable advice, we have implemented new techniques to handle potential noise samples that could negatively impact the VLM during annotation.
>
> 1. **Handling Stuck States Due to Launch and Network Issues**
>
> Temporary stalling or waiting periods are common in real-world mobile phone usage scenarios, often resulting from application launch delays, network timeouts, or system-level interruptions. These stuck states generate **trajectories containing redundant, meaningless screenshots or identical UI states** that do not contribute to the learning signal.
>
> To address this issue, we implement an automated detection mechanism that identifies consecutive identical or near-identical frames within a trajectory. Specifically, we calculate the structural similarity index (SSIM) between sequential screenshots to quantify visual similarity. When three or more consecutive samples exhibit a similarity threshold above 99%, we classify this segment as a stuck state and apply intelligent downsampling. That is, we retain only one representative sample from the sequence while preserving the trajectory's temporal fluency. This approach effectively removes noise while maintaining the essential flow of user interactions.
>
> 2. **Removing Repetitive Occurrences from User Navigation**
>
> Another prevalent source of noise in real-world interaction data stems from users' **repetitive navigation behaviors**, such as repeatedly going forward and backward through an interface or making accidental clicks that require correction. A particularly common pattern we observe is when users click on an unintended button and immediately navigate back, creating a sequence of: [State A] → [State B (wrong)] → [State A (back)] → [State C]. This pattern introduces unnecessary redundancy that can confuse our annotation process.
>
> We specifically target samples where identical images appear with one intermediate state in between (i.e., A-B-A patterns), which typically indicate user correction behaviors. When such patterns are detected, we prune the latter two images (the incorrect state and the return state) while preserving the original state and the final intended destination. This strategy effectively removes backtracking noise while maintaining the meaningful progression of user intent.
>
> &nbsp;
>
> ### Clarification of the Primary Problem and Future Directions
>
> 1. **Primary Problem of MobileA3gent**
>
> Also we would like to note that, the primary problem of this paper is **how to harness distributed user trajectory with minimal human intervention**.
> Since MobileA3gent is the initial work in the direction of user-centric GUI agent, our priority is to establish a foundation with normalized data before addressing the challenges associated with noisy data.
> We need to first answer the question, *How can we make this possible?* before delving into *How can we handle more noise types?*.
> We sincerely hope you may consider our major contribution, which includes both the proposal of the task and the complete architecture we have implemented.
>
> 2. **Future Directions for Other Noise Types**
>
> While we recognize that there are numerous sources of noise in real-world interaction data and that our current implementation may not address every possible noise pattern, our systematic approach to trajectory cleaning has demonstrated significant practical benefits. Our implementation effectively tackles the most common and impactful noise sources—stuck states and repetitive navigation patterns—which are prevalent across diverse mobile interaction scenarios.
>
> We are excited to see how the community will extend these techniques to address additional noise types and further advance the state of automated data curation for mobile interaction learning. Our work opens up promising avenues for future research, and we are confident that continued progress in this direction will unlock even greater potential for learning from real-world interaction data.
>
> ---

---

> ### Author Response · Authors · 2025-11-22
> **Response top Reviewer JzkA (3/3)**
>
> ## Response to Weakness 4: Efficiency Analysis
>
> > The paper omits a detailed analysis of client-side computation, communication frequency, or energy cost, which are critical for feasibility on mobile devices.
>
> We apologize for not including the efficiency analysis in the main body of the paper due to page limitations.
> To address your concerns, we present a detailed analysis from two perspectives: client-side computation and communication in MobileA3gent.
>
> &nbsp;
>
> ### Client-Side Computation
>
> We kindly note that we have provided detailed computational statistics, including memory usage and generation time (latency), in **Table 4, Appendix D.1**.
> For your convenience, we reproduce some of the results in Table A. The measurements are based on 1,000 samples. For users with only 50 trajectories, both time and cost are significantly reduced, **taking roughly one minute**.
>
> [**Table A**. Time and resource usage for both training and annotation.]
> |Model|Generation GPU (MB)| Generation Time/1000 Epi. (s)| Train GPU (MB)|Train Time/1000 Epi. (hours)|
> |--|--|-|-|-|
> |Phi-3.5-Vision-Instruct|9686|1374|11558|5.21|
> |InternVL2-1B|3985|1662|10330|3.15|
> |Qwen2-VL-2B-Instruct|12046|1180|15680|2.58|
>
> As shown in Table A, **deploying InternVL2-1B requires only 3GB of memory**, making it feasible for mobile devices equipped with 8GB RAM or more.
> Larger models, such as Phi-3.5 and Qwen2-VL-2B-Instruct, can also **fit within a 16 GB capacity**.
> Moreover, these requirements can be further reduced by adopting efficient training backbones.
>
> &nbsp;
>
> ### Communication
>
> 1. **Overhead**
>
> We assume all episodes have equal data size for approximation, and compare the communication overhead of three approaches:
> - Centralized training (one round of full dataset transmission)
> - Federated training with full fine-tuning (transmitting full model parameters for each round)
> - MobileA3gent, FL with LoRA (transmitting only LoRA adapters for each round)
>
> [**Table B**. Communication overhead using Qwen2-VL-7B on Android Control.]
> |Approach|Overhead|
> |--|--|
> |Central + Unpacked data (10,000 episodes)|≈100 GB|
> |Central + Unpacked data (1,000 episodes)|≈10 GB|
> |Central + Original TFRecord file (compressed)|50 GB|
> |FL + Full model (per round) |16.57 GB × round|
> |MobileA3gent, FL + LoRA adapter (rank=8, α=32, per round)| 77.06 MB × round|
>
> Table B demonstrates that LoRA-based FL is **the most communication-efficient method**. Note that the communication cost is identical across different federated algorithms used in our experiments.
>
> 2. **Frequency**
>
> Regarding communication frequency, as outlined in our vision for MobileA3gent (please refer to our response to Reviewer k9bV, question 2), we expect the user to communicate with the server on a daily basis, meaning **once per day**. Additionally, effective training, as demonstrated in our experiments, requires about only 10 communication rounds.
> Thus, the communication frequency is quite acceptable and will have minimal impact on the user's normal usage.
>
> ---
> &nbsp;
>
> ## Response to Question 1:
>
> > This dataset curation method might contain sparse and noisy data, i wonder how does it compare to centralized manual data curation
>
> Thank you for this question. Based on our understanding of it, you may be curious about how our Auto-Annotation method outperforms Human-Annotation.
> As elaborated in our experimental setup, since our curation method is significantly more cost-effective, we can easily acquire more data samples than manual curation while maintaining a tight budget. When trained with equal data size, our method experiences a modest performance drop (e.g. a 2% decrease in SR on AndroidControl-High). However, **as the data size scales up, our method surpasses Human-Annotation with ease while maintaining minimal cost**.
>
> Detailed experimental results are presented in **Table 5 and Appendix D.3**. For your convenience, we also include selected results in Table E.
>
> [**Table C**. Concise comparison between Auto-Annotation and Human-Annotation.]
> |Methodology|AndroidControl-High|||AndroidControl-Low|||GUI Odyssey|||
> |--|-|-|-|-|--|-|--|-|-|
> ||Type|Ground|SR|Type|Ground|SR|Type|Ground|SR|
> |DataSize=5000/3000||||
> |**Human-Annotation**|79.14|66.56|61.70|97.62|81.47|85.99|84.39|75.63|67.01|
> |**Self-Instruct**|75.86|57.28|53.95|97.47|81.97|85.25|82.80|60.27|55.16|
> |**Auto-Annotation**|77.49|62.67|58.12|98.06|83.29|86.29|81.72|69.51|60.57|
> |DataSize=1000|
> |**Human-Annotation**|75.41|53.75|50.97|97.02|74.66|80.48|78.85|64.92|55.22|
> |**Self-Instruct**|72.43|48.99|47.54|96.87|72.40|78.69|77.07|51.33|45.22|
> |**Auto-Annotation**|74.22|52.44|49.48|97.47|75.13|80.48|76.56|53.40|46.37|
>
> ---
> &nbsp;
>
> Overall, following your suggestions, we have made specific revisions including denoising techniques, privacy analysis and statistics on computation & communication. We hope our responses adequately address your concerns.
> We appreciate your feedback and will be grateful if you might consider increasing the score.

---

> ### Author Response · Authors · 2025-11-27
> **Kind Reminder**
>
> Dear Reviewer JzkA,
>
> We gently invite you to review our additional experiments, statistics, privacy analysis and points of clarification, as your feedback is invaluable for strengthening our work. We hope that we have sufficiently addressed your concerns, and if there remain any further questions we would be more than happy to provide additional explanations.
> We appreciate your time and consideration,
>
> The Authors

---

### Official Review · Reviewer_k9bV · 2025-11-01

**Soundness:** 2
**Presentation:** 2
**Contribution:** 2
**Rating:** 2
**Confidence:** 4

**Summary:**

This paper introduces MobileA3gent, a framework for training mobile GUI agents using decentralized user data while preserving privacy. It features Auto-Annotation (automatic data collection from daily phone usage via local VLMs, reducing costs by 99%) and FedVLM-A (federated learning with adapted aggregation for mobile data heterogeneity). Experiments show that this approach achieves performance comparable to manual annotation at only 1% of the cost across multiple benchmarks.

**Strengths:**

1. Interesting topic of mobile GUI agents
2. Good comparison with existing baselines
3. Good quality of figures, use of font sizes etc.
4. Appreciate the open sourcing of the code.

**Weaknesses:**

1. Overall the paper was not an easy read. While there are no spelling/grammar issues, I took me a while to understand the motivation of this work, how they collect the data, where the data are processed/annotated through the VLM, the type of data that are collected etc. I strongly recommend the authors to revise the manuscript and provide clear examples of data samples, and system design diagram of the system's pipeline. I would also avoid the inline tables and figures (just a friendly suggestion) as it makes the manuscript hard to read.
2. My biggest concern is the contribution of this work. While the area of mobile GUI agents is very important, I find the approach impractical. The authors suggest that an on-device VLM that constantly decodes and annotates screenshots is something that can be mainstream in large studies. However, VLMs are extremely expensive, only available in high-end devices and consume extreme device power. Even assuming that things will change in the future, I don't expect that it will be realistic to use VLM in the background while a user accesses his/her device. Moreover, use of VLMs will introduce additional UX issues (latency, device high temperature, etc.). Offline analysis could be possible, but you would need to deal with many existing mobile OS limitations (in both Android and iOS).
3. The authors claim that the system is "preserving privacy" of the users, however there is no privacy analysis, state of thread model etc. besides a small paragraph in L264.

**Questions:**

1. Could the authors provide a more detailed description of the data collection and annotation process to clarify how the data flow through the system and what exactly the model sees during training and inference?
2. How do the authors envision this approach being deployed in real-world large-scale studies given current hardware limitations?
3. Could the authors elaborate on the specific privacy guarantees of the system, including what data are exposed or stored, how they are protected, and what assumptions are made about adversaries?

---

> ### Author Response · Authors · 2025-11-22
> **Response to Reviewer k9bV (1/3)**
>
> Thank you for the time and effort you dedicated to reviewing our paper. With your expertise, we believe we can make further progress toward a more cost-effective and scalable solution.
> Below, we address each of your concerns accordingly.
>
> ---
> &nbsp;
>
> ## Response to Weakness 1: Paper Writing
> > Overall the paper was not an easy read. While there are no spelling/grammar issues, I took me a while to understand the motivation of this work...
>
> We sincerely apologize for the difficulty you experienced in reading the manuscript. In the revised version, we have improved the clarity of the motivation, data collection process, and data examples by refining the narrative and highlighting key sentences in bold for better readability.
>
> > I strongly recommend the authors to revise the manuscript and provide clear examples of data samples, and system design diagram of the system's pipeline.
>
> Thank you for your constructive recommendation. We have already included a representative data example in **Figure 11 (page 23)**. We apologize for not placing this example in the main text due to the page limit, and we will move it into the main body if we are granted an additional page.
> Regarding the system design, **Figure 2** provides a diagram of the overall pipeline. We warmly welcome any suggestions for improving this figure, and we would be happy to refine it further.
>
> As we are keen to improve the quality of our manuscript, we would greatly appreciate any more specific suggestions on writing or presentation. We are more than willing to incorporate them and further revise the paper.
>
> ---
> &nbsp;
>
> ## Response to Weakness 2: Practical Concern & Resources
>
> > VLMs are extremely expensive, only available in high-end devices and consume extreme device power. Even assuming that things will change in the future, I don't expect that it will be realistic to use VLM in the background while a user accesses his/her device.
>
> Thank you for raising this point.
> We address this concern by elaborating on the feasibility of on-device VLM deployment from three perspectives:
>
> &nbsp;
>
> ### Time and Resource Statistics for Annotation and Training
>
> We provide detailed resource statistics, including memory usage and generation time (latency), in **Table 4, Appendix D.1**.
> For your convenience, we reproduce the results in Table A. The measurements are based on 1,000 episodes.
> For users with only 50 trajectories, both time and cost are significantly reduced, **taking roughly one minute**.
>
> [**Table A**. Time and resource usage for both training and annotation.]
> |Model|Generation GPU (MB)|Generation Time/1000 Epi. (s)|Train GPU (MB)| Train Time/1000 Epi. (h)|
> |-|-|-|-|-|
> |Phi-3.5-Vision-Instruct|9686|1374|11558|5.21|
> |InternVL2-1B|3985|1662|10330|3.15|
> |Qwen2-VL-2B-Instruct|12046|1180|15680|2.58|
>
> Notably, Intern2-VL-1B has the lowest GPU requirements, needing only 4GB for generation and 10GB for training. This makes it **feasible for many mobile devices with 16GB of RAM or more**.
> Larger models, such as Phi-3.5 and Qwen2-VL-2B, can also **fit within a 16 GB capacity**.
> Moreover, these requirements can be further reduced by adopting efficient training backbones.
>
> &nbsp;
>
> ### On-Device Deployability Supported by Substantial Research
>
> 1. Importantly, according to many technical reports (e.g. Phi models [1], MobileAgent-V3 [2] and AppCopilot [3]), the relative small VLM models including *Phi-3.5 and InternVL2-1/2B*, are actually **deployable on mobile devices**.
> 2. Moreover, an increasing number of studies have focused on developing lightweight models tailored for mobile tasks, such as LiMAC [4] and AppVLM [5]. These efforts toward building smaller, more efficient VLMs further support the practical viability of on-device GUI agents.
>
> &nbsp;
>
> ### Practical Usage Considerations
>
> We understand your concern about running VLMs in the background while a user is actively using their device, as this can certainly impact memory and battery usage. However, we view this as primarily a usage consideration, rather than a deployment limitation. MobileAgent does not need to operate concurrently with all user interactions.
> For example, we can record user trajectories only if the user opts in for collaboration. The data can then be **processed asynchronously when the device is idle, such as during sleep hours**, thus minimizing any impact on device performance.
>
> &nbsp;
>
> ### Summary
>
> In conclusion, the deployment of VLMs on mobile devices is increasingly feasible, driven by advancements in model efficiency and growing support for lightweight architectures. While there are legitimate concerns regarding concurrent usage, these challenges can be mitigated through careful deployment strategies, such as asynchronous processing and selective model activation.
> Therefore, based on our detailed resource statistics, we believe that the use of VLMs in real-world mobile environments is not only possible but also practical for a broad range of use cases.

---

> ### Author Response · Authors · 2025-11-22
> **Response to Reviewer k9bV (2/3)**
>
> ## Response to Weakness 3 and Question 3: Privacy Analysis
>
> > W3: The authors claim that the system is "preserving privacy" of the users, however there is no privacy analysis, state of thread model etc. besides a small paragraph in L264.
>
> > Q3: Could the authors elaborate on the specific privacy guarantees of the system, including what data are exposed or stored, how they are protected, and what assumptions are made about adversaries?
>
> Thank you for raising this important point. We recognize that the original submission could not provide a detailed discussion on privacy due to page limitations.
> In the following revision, we have added a dedicated and more comprehensive privacy analysis, covering:
> - the threat model and assumptions regarding potential adversaries,
> - what data are stored or transmitted within the system, and
> - the corresponding privacy protection mechanisms.
> These additions clarify the specific privacy guarantees offered by our system and we hope they address the concern raised in the review.
>
> &nbsp;
>
> ### Threat Model
>
> **1.1 Adversaries**
>
> We consider multiple potential adversaries:
> - External Eavesdroppers: Attempt to intercept communication between client and server.
> - Honest-but-Curious Server: Follows the training protocol but tries to infer private user information from uploaded model updates.
> - Data-Abusive Server: May attempt to store, reuse, or monetize uploaded data for secondary purposes.
> - Curious Peer Clients: Aim to extract information about other users by inspecting shared data.
>
> **1.2 Adversary Capabilities**
>
> We assume adversaries may:
> - Observe and record communication traffic (for external eavesdroppers).
> - Inspect uploaded information to perform gradient inversion attacks in an attempt to infer screenshots, actions, or user intents (honest-but-curious servers).
> - Store or repurpose any received data (for data-abusive servers), including attempting to build centralized datasets for secondary use.
> - Infer other users’ information, aiming to identify patterns attributable to specific clients (curious peer clients).
>
> **1.3 Trust Assumptions**
>
> - Client devices behave correctly and do not intentionally exfiltrate data beyond the MobileA3gent protocol.
> - The server does not actively tamper with models but may analyze updates (honest-but-curious).
> - Communication channels may be passively monitored but remain encrypted (e.g., TLS).
> - Attackers cannot directly access raw screenshots or operation logs stored on devices.
>
> &nbsp;
>
> ### Data Exposure and Storage
>
> **2.1 Data Stored Exclusively on User Devices**
>
> MobileA3gent ensures that the following remain strictly local and never transmitted:
> - Raw screenshots and actions
> - Intermediate artifacts generated by Auto-Annotation
> - Locally constructed training episodes.
>
> **2.2 Transmitted Data**
>
> The only information transmitted from clients to the server is:
> - LoRA parameters. These updates contain no raw user inputs and are highly compressed.
>
> &nbsp;
>
> ### Privacy Guarantees
>
> **3.1 Local Data Retention**
>
> The key privacy guarantee arises from fully localizing data processing.
> No raw visual or behavioral data ever leaves the device.
> ```
> Guarantee 1 – No raw data leakage
> Sensitive GUI content, user behavior logs, and annotated intent signals remain entirely on the user's device.
> ```
>
> **3.2 Resistance to External Eavesdropping**
>
> Since clients only transmit encrypted model updates:
> - External adversaries cannot observe screenshots or user actions.
> - Encrypted model updates make it difficult to reconstruct private data.
> ```
> Guarantee 2 – Secure communication
> Network listeners cannot directly infer any application content, or personal information.
> ```
>
> **3.3 Server-Side Protection**
>
> The server only receives:
> - Sparse LoRA updates
> - Non-sensitive metadata (episode and step counts)
>
> It never observes raw screenshots, actions, or instruction texts, and the aggregation procedure relies solely on statistical signals.
> Due to the lack of raw visual data, gradient inversion becomes significantly harder.
> ```
> Guarantee 3 - Limited server inference
> The server cannot access screenshots, user intents, or application states.
> ```
>
> Since no content-level data is ever uploaded, the server provider cannot repurpose or resell user data.
> No centralized dataset exists to be exploited or leaked.
> ```
> Guarantee 4 — Prevention of data abuse
> The server receives no reusable or sensitive data, eliminating risks of secondary use or unauthorized dataset accumulation.
> ```
>
> **3.4 Protection against Curious Peer Clients**
>
> Clients never communicate with each other directly.
> All cross-client information is aggregated by the server, removing client-specific details.
> ```
> Guarantee 5 – Peer isolation
> No client can access or infer any other user’s data or model updates.
> ```
>
> &nbsp;
>
> We hope this revised privacy analysis adequately addresses your concern.

---

> ### Author Response · Authors · 2025-11-22
> **Response to Reviewer k9bV (3/3)**
>
> ## Response to Question 3: Vision of MobileA3gent
>
> > How do the authors envision this approach being deployed in real-world large-scale studies given current hardware limitations?
>
> To answer your third question, our vision for MobileA3gent in the future is as follows:
> 1. The user can choose to participate in the collaboration by contributing a trajectory in exchange for something, such as a small monetary reward, which is far less than the cost of hiring a professional annotator, or free access to our advanced GUI agent.
> 2. The background program only needs to record the trajectory and temporarily store the data on the device.
> 3. During idle time, such as sleeping hours, the framework will perform annotation and training using lightweight models, typically smaller than 4B.
> 4. The server will asynchronously aggregate updates from different users, resulting in a stronger GUI agent.
> 5. The enhanced model can then encourage more users to participate and contribute data.
>
> This approach offers tremendous scalability and cost-efficiency. As more users participate and contribute their data, the system's performance continuously improves, creating **a snowball effect**. With each new contribution, the system becomes more robust, **attracting even more users to join the collaboration**.
> This cycle fosters exponential growth, where the benefits increase as the user base expands, ultimately leading to a sustainable and highly efficient GUI agent for on-device training and annotation.
>
> ---
> &nbsp;
>
> ## Reference
>
> [1] Phi-3 Technical Report: A Highly Capable Language Model Locally on Your Phone. Microsoft, 2024.
> [2] Mobile-Agent-v3: Fundamental Agents for GUI Automation, 2025.
> [3] AppCopilot: Toward General, Accurate, Long-Horizon, and Efficient Mobile Agent, 2025.
> [4] Lightweight Neural App Control. ICLR 2025.
> [5] AppVLM: A Lightweight Vision Language Model for Online App Control. 2025.
>
> ---
> &nbsp;
>
> Overall, we hope have successfully addressed you concerns and questions by improving the writing and providing comprehensive privacy analyses. If you have further suggestions, we are more than pleased to adopt them and improve this paper.
> We appreciate your feedback and will be grateful if you might consider increasing the score.

---

> ### Author Response · Authors · 2025-11-27
> **Kind Reminder**
>
> Dear Reviewer k9bV,
>
> Thank you for your constructive feedback on our paper. With the discussion phase ending shortly, we gently invite you to review our responses. We hope that we have sufficiently addressed your concerns, and if there remain any further questions we would be more than happy to provide additional explanations.
>
> We sincerely appreciate your time and consideration,
>
> The Authors

---

### Author Response · Authors · 2025-12-03
**Summary of Rebuttal for AC**

We sincerely thank all reviewers and the committee for their time and effort.
In light of the recent policy change, we summarize below the core contributions of our work and how our responses and additional experiments address the main concerns.

---
&nbsp;

## Summary of Strengths and Contributions

1. Addresses a critical and timely research problem.
> Practical and meaningful research problems (QgZJ), a pressing issue in the GUI agent community (JzkA), Interesting topic (k9bV)
2. Proposes a cost-efficient and scalable agent framwork with two novel techniques.
> Auto-Annotation is conceptually novel (JzKA), Practical FL innovation (ybJy)
3. Conduncts extensive experiments demonstrating strong effectiveness and potential.
>  technically sound and thoroughly validated (ybJy), Extensive experiments (QgZJ), extensive and rigorous (JzkA), Good comparison (k9bV)
4. Presents a well-organized and clearly written paper with high-quality figures.
> well-structured and easy to follow (JzkA, ybJy, QgZJ), Good quality of figures (k9bV, ybJy)

We believe this research not only establishes important groundwork for a novel distributed user-centric approach, but also opens promising avenues for future studies, including handling heterogeneity, reinforcement learning, and more.

---
&nbsp;

## Addressing Concern 1: Practical Feasibility and Efficiency Analysis

> k9bV W2, JzkA W4, ybJy W1 and W2, QgZJ W1.

1. We added **detailed computation and communication statistics** for both generation and training, **reported in Tables 4, 5, and 6** of the revised paper (or Tables A, B, and D in the response document).
The results show that models such as Phi-3.5, Qwen2-2B, and InternVL2-1/2B can **fit within phones equipped with 16GB RAM**, which is common in today’s mobile market.
2. Our extensive model-wide experiments (Figures 6 and 9) further demonstrate the **model-agnostic nature** of MobileA3gent and its robust compatibility with diverse models.
3. The practicality of on-device GUI agents is further reinforced by **a growing body of related work** (e.g., LiMAC, Mobile-Agent-V3) and accelerated by emerging industry deployments (e.g., the recently released **Doubao Mobile Assistant**).

We therefore believe on-device GUI agents are *both feasible and an inevitable future trend*.
```
Details available in our response to Reviewer ybJy (1/2) and (2/2).
```

---
&nbsp;

## Addressing Concern 2: Privacy Analysis

> k9bV W3, JzkA W2

We have added **a dedicated and substantially expanded privacy analysis**, covering:
- threat model and assumptions regarding potential adversaries,
- data stored or transmitted within the system, and
- corresponding privacy-preserving mechanisms.

This improved privacy analysis has been incorporated into Section 3.4 of the revised paper.
```
Details available in our response to Reviewer k9bV (2/3).
```

---
&nbsp;

## Addressing Concern 3: Annotation Noise

> JzkA W3, QgZJ W3

We have implemented **new techniques to handle potential noise samples** that could negatively impact the VLM during annotation.
Our implementation effectively tackles the most common and impactful noise sources—stuck states and repetitive navigation patterns—which are prevalent across diverse mobile interaction scenarios.
```
Details available in our response to Reviewer JzkA (2/3).
```

---
&nbsp;

## Addressing Concern 4: Real-World Experiments & On-Device Data Collection

> QgZJ W4, JzkA W1

We have added extensive experiments to improve the realism and comprehensiveness of our evaluation.
**Tables E, F, G, and H present new results focusing on realistic simulation, client scaling, data heterogeneity, and out-of-distribution generalization.**
```
Details available in our response to Reviewer QgZJ (3/4) and (4/4).
```

Regarding on-device data collection, we note (as discussed in Section B.2) that directly collecting user data raises significant **ethical and logistical challenges**. While open-source datasets enable fair comparison and public release, they may not fully reflect real-world usage. We therefore adopt the latter approach to advance research while upholding ethical standards.

```
Details available in our response to Reviewer JzkA (1/3).
```

---
&nbsp;

We hope this concise summary assists the area chair in evaluating the core contributions of MobileA3gent, and demonstrates that the concerns raised in the reviews have been thoroughly addressed in both the revised manuscript and our responses.
Thank you again for your valuable time and contributions to the community.

---

### Meta-Review · Area_Chair_UALT · 2026-01-07

**Summary:**

The reviewers primarily focused on the practicality and real-world validation of the proposed framework. Key concerns included:



* Feasibility on Mobile Hardware: Multiple reviewers (k9bV, JzkA, ybjy, QgZJ) questioned whether running memory-intensive VLMs for both annotation and training is realistic on modern smartphones due to battery, heat, and RAM constraints.


* Privacy Analysis: Reviewers k9bV and JzkA noted the lack of a formal threat model or deep analysis of how the system protects against adversarial attacks on decentralized data.

* Annotation Noise: Concern was raised regarding how "auto-annotation" handles errors (e.g., stuck states or accidental clicks) and if these errors propagate to degrade the final model.

* Simulation vs. Real World: All experiments used publicly available datasets rather than actual on-device collection, leading to questions about the practicality of the findings.

**Reviewer Concerns:**

Addressed by Rebuttal

* Hardware Feasibility: The authors provided detailed resource statistics (Tables A & D) showing that models like InternVL2-1B can operate within 4GB to 10GB of RAM, making them compatible with 16GB RAM devices. They also suggested asynchronous processing during idle "sleep hours" to avoid impacting active usage.

* Privacy Framework: The authors added a comprehensive threat model identifying four types of adversaries (e.g., "honest-but-curious" servers) and outlined five specific privacy guarantees. They also implemented secure aggregation to protect gradient updates.

* Denoising Techniques: New automated mechanisms were implemented to handle stuck states (using SSIM index) and repetitive navigation (A-B-A patterns) to clean noisy data before training.

* Communication Efficiency: The authors demonstrated that transmitting LoRA adapters (approx. 77MB) is vastly more efficient than transmitting full models or raw data.

Still Outstanding Concerns

* Empirical On-Device Validation: While the authors provided cloud-based simulations, they admitted that running actual on-device experiments remains difficult due to current hardware/logistical constraints. The gap between simulation and true deployment remains unverified by physical testing.


* Real-World User Interaction: The datasets used are still pre-existing public benchmarks. The variability and "messiness" of true daily phone usage (unrelated to the training tasks) have not yet been captured in their experiments.

* Algorithmic Innovation: Since it is not a systems paper (simulations on public datasets), it is not clear what are the algorithmic contributions (adapted aggregation has appeared in prior works).

**Reviewer Scores:**

* k9bV: Mixed. The author's extensive privacy analysis and the inclusion of a system design diagram can address this reviewer's concern about "reading difficulty" and "missing privacy analysis" but not completely clear.
+2

* JzkA: Mixed. The new denoising techniques and detailed communication statistics specifically addressed their questions on noise and efficiency.

* ybjy: Already positive, this reviewer asked for specific VRAM/latency stats for edge hardware. The authors provided these exact figures in Table D, reinforcing the reviewer's view that the work is "technically sound".

* QgZJ: Positive. The authors provided the experiments (100 clients, 60,000 steps) that this reviewer explicitly requested in Question 2.

---

### Decision · Program_Chairs · 2026-01-26

Reject